# Checklist of Macrofungi Associated with Nine Different Habitats of Taburno-Camposauro Massif in Campania, Southern Italy

**DOI:** 10.3390/jof10040275

**Published:** 2024-04-09

**Authors:** Maurizio Zotti

**Affiliations:** Department of Agricultural Sciences, University of Naples Federico II, Via Università, 100, 80055 Portici, Italy; maurizio.zotti@unina.it

**Keywords:** Basidiomycota, Ascomycota, Gadgil effect, mycobiome, bioindicator species

## Abstract

The checklist serves as an informative method for evaluating the diversity, geography, and ecology of established and reproducing macrofungi. Additionally, considering macrofungi as bioindicator species, their census should be incorporated into efforts to monitor the state of health of ecosystems and directly applied to conservation policies. Between 2019 and 2023, a census of macrofungal species was conducted in Taburno-Camposauro Regional Park (Campania, Italy) across nine distinct habitats. A total of 453 fungal taxa were identified, including several new records for the Campania region. The fungal diversity exhibited significant variations based on the dominant plant species in each habitat. Fagacean tree species and *Carpinus* spp. shared similar fungal communities. Equally, coniferous tree species displayed a comparable fungal composition. In *Abies alba* and mixed broad-leaved forests, low levels of ectomycorrhizal taxa were observed alongside a concurrent increase in saprotrophs, indicating a disturbed habitat and a reduction in the Gadgil effect. Notably, lower fungal diversity was documented in the grassland habitat, suggesting the potential implications of wildlife imbalance and excessive grazing. The provided checklist constitutes a valuable resource for local management authorities, providing insights to formulate specific management policies.

## 1. Introduction

Fungi are one of the most complex and cryptic kingdoms on Earth. Their biology, ecology, and distribution are still not well understood. The diversity of this wide kingdom is markedly underestimated and deserving of a massive research effort [1,2]. Within the vastity of the fungal kingdom, a convenient opportunity to study and give a glimpse into their geographic distribution and their organization within hosting habitats comes from territorial census and checklist production of macrofungi [3,4,5]. Nowadays, fungal checklists are used to assess biological diversity in a specific geographic area [6,7], to assess the ecological relationship between fungi and plant communities [8,9,10]; additionally, checklists are helpful means to categorize fungal taxa as bioindicator species, providing helpful information in light of macroecological effects due to territorial management policies [11]. For this reason, macromycetes census, despite representing one of the oldest methods in mycology, remains an actual and representative practice to assess the presence of a reproducing population of fungi in a specified environment [1,5,12]. More recently, ITS-based metagenomics techniques can also provide comprehensive information on fungal communities. However, these methodologies, despite being useful to assess the quantitative relationships of microbial abundance with environmental variables [13], cannot be a substitute for checklist approaches, as the morphological identification of fungal sporophores indicates the presence of an established and reproductive fungal species. [13,14,15].

In 2005, a national overview of fungal diversity of Italy was presented in a comprehensive checklist of fungi [12]. The work gathered information from previous studies in the Italian peninsula and aimed to fill the gap of knowledge regarding fungal distribution in Europe [16]. A lack of updated data was evidenced for Italy, where macrofungi were understudied, despite the Italian school of mycology historically being considered one of the most representative at a global level. To fill the gap, the Italian checklist of fungi in 2005, made on regional basis, provided a comprehensive assessment of the diversity of fungi in Italy [17,18]; The authors of the work stated that in some regions fungal diversity was underestimated, probably because of the lack in mycologist and/or mycological associations involved in census projects [12]. Hence, Molise and Basilicata, regions that are well known for their extensive natural environments and woodland habitats, were less studied compared to the northern regions of Tuscany, Lombardy, and Veneto that were well represented because of the presence of well-established schools of mycology and mycological organizations. Sicily, Calabria, and Sardinia were also well represented because of a remarkable activity of mycological groups and scientists as well as a particularly diversified flora [19].

In the Campania region, fungal diversity was poorly studied; although it has an important mycological history and a consistent heterogeneity of habitats, only 658 macromycetes were listed in 16 territorial groups divided into three islands, eight mountain chains, three volcanic areas, one botanical garden and one coastal environment [20]. Even the authors of the latest checklist of the region stated that the census produced does not constitute a definitive document because of the limitations of covering the complex and mosaic-like distribution of habitats in Campania. Hence, attention should be focused on gathering information from several small territorial conservation areas, as reduced territorial extension allows a reasonable data cover without excessive collection effort. Considering this, in the present work, by using checklist methodology and periodic sampling, attention was focused on the Taburno-Camposauro massif, located in the Benevento province of Campania, called “La Dormiente del Sannio”. The massif was included in the study of Violante et al. 2002 [20] and was subjected to a conservation project between 2019 and 2023 that also focused on the aspect of macrofungal occurrence (Project name: “Sve(g)liamo la Dormiente”, financed by “Fondazione con il Sud”). To date, the Taburno-Camposauro massif, and all the territories included in the regional park, hosts a considerable diversity of habitats because of its geographic location, human activities, and stringent conservation policies [21,22,23]. These characteristics make this area, and other regional parks in Campania, a good study site that could potentially host many niches for diverse fungal taxa, helping to assess macrofungal diversity in Mediterranean areas.

## 2. Materials and Methods

### 2.1. Study-Site Description and Vegetation Types

This study was conducted in the territory of Taburno-Camposauro Regional Park in Campania, Italy (established in 1994). The area comprises approximately 280 km^2^. The regional park includes the Taburno-Camposauro massif that is part of the Italian Apennines chain and constitutes a geologically distinct mountain range with an altitude ranging from 400 to 1394 m a.s.l. (peak of mount Taburno), characterized by limestone formations, rocky cliffs, and valleys. Taburno-Camposauro massif is flanked by two neighboring mountain ranges. To the north lies the Matese massif and to the south the Partenio massif. The Taburno-Camposauro massif is positioned at approximately 41.0760° N latitude and 14.8456° E longitude. The area experiences a Mediterranean climate with features that fluctuate according to altitude. From its foothills to its peaks, the Taburno-Camposauro massif showed different vegetation planes, distinguished in belts with a diverse organization of deciduous forests, followed by conifers and Mediterranean grasslands at medium-higher elevations. The study areas were selected according to vegetation types, dominant plant species, and the presence of Natura 2000 conservation sites, identified according to the manual for the interpretation of habitats of directive 92/43/EEC [24]. In detail, the habitats that were the subjects of this study are showed in Figure 1 and categorized according to the “nature maps of Campania region” available at https://www.isprambiente.gov.it/it/servizi/sistema-carta-della-natura/cartografia/carta-della-natura-alla-scala-1-50.000/campania (accessed on 15 December 2023) [25,26]. Habitats considered in the present study are:Forests of *Fagus sylvatica*, encompassing around 21.15 km^2^ of regional park land and constituting the highest vegetation belt, with a minimum elevation of 800 m a.s.l. In the Taburno-Camposauro massif, the *F. sylvatica* forests include the habitats 9210 “Apennine beech forests with *Abies alba*” (Taburno) and 9220 “Apennine beech forests with Taxus and Ilex” (Camposauro). Both are considered priority habitats.Mixed forests with a prevalence of *Carpinus* spp. That encompass 48.80 km^2^ of land and are considered the most representative vegetation belt in the massif ranging from piedmont to 1000 m a.s.l. The area includes the habitat 91L0, corresponding to “Illyrian oak-hornbeam forests” and characterized by the cooccurrence of several deciduous tree species alongside the *Carpinus* spp., *Fraxinus*, *Acer*, and *Quercus* tree species; they cooccur with a scattered distribution.Grasslands of around 13.4 km^2^. This habitat type encompasses semi-natural dry grasslands that develop on calcareous (lime-rich) substrates at an elevation high above the woody vegetation and managed lowland prairies. This habitat is a proritary one and is mainly recognized in Natura 2000 as 6210, “Semi-natural dry grasslands and scrubland facies on calcareous substrates (Festuco-Brometalia)”.Woodlands of *Castanea sativa*, (7.1 km^2^) occupying the foothills vegetation belt and strongly favored by humans for chestnuts and wood production. The Natura 2000 habitat within the area is recognized as 9260, “*Castanea sativa* woods”.*Quercus pubescens* forests, encompassing 3.71 km^2^ of land, are represented by patches of vegetation that are mostly localized in arid south-facing sloped areas favoring thermophilus vegetation. In that area, the priority habitat 91AA is observed and described as “wooded areas dominated by various species of white oaks”.Allochthonous conifers refers to a reforested area with mostly two-needle pine trees (*Pinus nigra*, *P. pinaster*, *P. halepensis.*, *P. pinea*) but also several *Cupressus* species, *Pseudotsuga menziesi* and *Larix* spp. This category is present in the park as result of reforestation policies enacted in the area in the 1970s. Those habitats are distributed at different altitudes within the park area. The actual surface of this category is around 1.38 km^2^ and includes the 9540 habitat, “Mediterranean formations dominated by Aleppo pine”.Forests of *Quercus cerris* account for a surface area of around 0.48 km^2^ and are mostly found in the form of monodominated patches in mesophilic environments. The Natura 2000 habitat within the area is recognized as 91M0 and defined as “Pannonian-Balkanic turkey oak-sessile oak forests”.The forest of *Abies alba* is localized in mount Taburno and measures approximately 0.42 km^2^. The area hosts the priority habitat 9510 “Southern Apennine silver fir forests”.Mixed broad-leaved forests are considered areas with a presence of several tree species, mostly riparian, and cover 0.03 km^2^. Those areas are identified as small patches of vegetation in valleys and along water bodies and are mainly dominated by the presence of *Salix* spp., *Populus* spp. mixed with *Carpinus* spp. *Quercus* spp., *Acer* spp. *and Fraxinus* spp. In the area, habitat 92A0 and 91L0, known as “*Salix alba* and *Populus alba* galleries” and “Illyrian oak-hornbeam forests” are present.

**Figure 1 jof-10-00275-f001:**
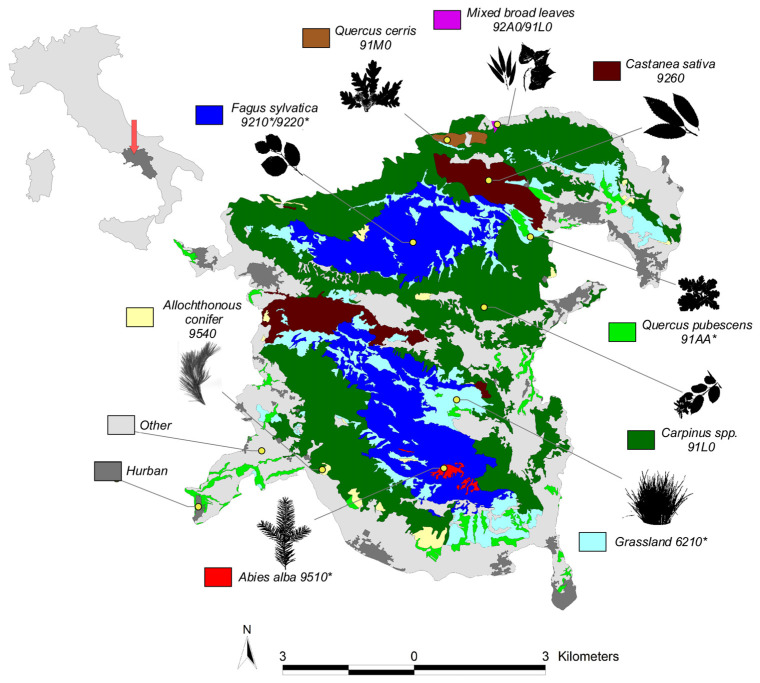
Maps of Taburno-Camposauro regional park (Campania) and position according to Italian geographic map (red arrow). Different colors indicate vegetation types according to nature maps of Campania region available at https://www.isprambiente.gov.it/it/servizi/sistema-carta-della-natura/cartografia/carta-della-natura-alla-scala-1-50.000/campania (accessed on 15 December 2023) and related Natura 2000 habitats classifications observed in the mycological survey. Asterisks in habitat code indicate priority habitats according to Natura 2000 classification.

### 2.2. Sampling Methodology and Taxa Determination

From 2019 to 2023, sampling was conducted in different habitats of Taburno-Camposauro Regional Park and was concentrated from the end of spring until winter. In these periods, normally lasting from April to November, sampling was conducted fortnightly or weekly according to climatic conditions favoring the emergence of fungal sporophores. For hypogeum taxa as *Tuber* spp., sampling was conducted with help of local truffle hunters. Some records were provided from the mycological inspectorate of ASL “Azienda Sanitaria Locale” of Benevento.

For each collection, samples were transported to the Laboratory of the Orazio Comes library in MUSA (Department of Agricultural Sciences of Federico II University of Naples, Portici) for species identification. Specimens were determined according to sporophore morphologic features and a microscopic analysis was performed using a similar approach to that reported by Ferraro in 2022 [5].

Ecological aspects of each species were assessed by comparing genus and species with databases of fungal trophic strategies FUNGUILD and comprehensive reviews on mycorrhizal-plant associations [27,28,29]. Nomenclature was readapted to those adopted in 2022 by Ferraro [5] as: Ecm = Ectomycorrhizal, Pm = Parasite on mushrooms, Pn = Necrotroph parasites, Sbg = Saprotrophs on burnt ground, Sd = Saprotrophs on dung, Sc = Saprotrophs on cones, Scl = Saprotrophs on cladodes, Scu = Saprotrophs on cupules, Se = Saprotrophs on exuviate, Sh = Saprotrophs on humus, Sl = Saprotrophs on litter, Sle = Saprotrophson leaves, Sm = Saprotrophs on mosses, St = Terricolous saprotrophs, Sw = Saprotrophs on wood, and UNK = Unknown.

Once species were determined, names were checked for synonymy in Index fungorum (https://www.indexfungorum.org/names/Names.asp, accessed on 10 February 2024). Part of the samples were dried and conserved in the mycological herbarium of the MUSA (Centro Musei delle Scienze Agrarie) in the Department of Agriculture Science of Federico II University of Naples.

### 2.3. Map Production and Data Analysis

The vegetation map in Figure 1 was produced in ArcGIS software (version 10.5) by overlapping vegetation layers of nature maps in the Campania region and a shape file of Taburno-Camposauro Regional Park. Some categories were grouped to match the vegetation classification recognized during field survey. Agricultural systems and urban centers were unified and shown in grey as they were not subject to the study.

According to geographic habitat grouping, the resulting dataset consisted of a presence/absence matrix of fungal taxa divided by fungal families, order, and trophic mode (ecological categories). For each specified category, data were cumulated to obtain frequency tables. Those data were transformed in relative abundance by the total of observations within the habitat and are shown in stacked bar plots. To assess the similarity in macrofungal composition among fungal families, order and trophic mode associated with different plant species and environments, habitat categories were ordered according to dendrograms based on a contingency matrix of Bray–Curtis similarity values. Dendrograms were built according to a complete linkage algorithm. Data analyses were conducted in Primer 7 software.

## 3. Results

The survey of macrofungal taxa in the Taburno-Camposauro massif identified a total 453 taxa. Of those, 257 had already been recorded in Campania, while 195 represented new records (Figure 2A). Ninety-six percent of the species observed belongs to Basidiomycota phylum while Ascomycota accounts for around 4% of the total (Figure 2B). The habitat with highest number of species was *F. sylvatica* 9210*/9220* (202, taxa), followed by *Q. cerris* 91M0 (177), *C. sativa* 9260 (132), mixed broad-leaved 92A0/91L0 and *Q. pubescens* 91AA* (101 taxa each), Allochthonous conifer 9540 (81), *A. alba* 9510* (80), *Carpinus* spp. 91L0 (75), and Grassland 6210* (43) (Figure 2C).

The most frequent order in the Taburno-Camposauro massif was represented by taxa belonging to Agaricales (49.95%), followed by *Boletales* (15.21%), *Russulales* (11.28%), *Polyporales* (7.55%), *Phallales* (4.73%), *Cantharellales* (4.13%), *Pezizales* (3.02), and *Lycoperdales* (1.31%). *Dacrymicetales*, *Helotiales*, *Nidulariales*, *Thelephorales*, *Xilariales*, and *Tremellales* were present with values below 1% of the total. The abundance of fungal orders, divided by habitats, is shown in Figure 3. Grassland fungal flora is the less similar compared to other habitats, with Agaricales accounting for 88% of the observations, followed by *Lycoperdales* (~10%). Differentiation of grassland habitats was due to the total absence of *Boletales*, *Russulales*, and *Cantharellales* that were always present within different forest habitats. *A. alba* showed a higher level of similarity with fungal flora of allochthonous conifers; the two (conifer) categories clearly segregated with respect to the other six categories mostly because of the higher occurrence of *Russulales*, *Polyporales*, and *Phallales*. A different organization of community was observed for mixed broad-leaved forests with respect to *C. sativa*, *Q. cerris*, *Q. pubescens*, *Carpinus* spp., and *F. sylvatica.* The first showed a higher percentage of Agaricales and a lower percentage of *Russulales*, while, *C. sativa*, *Q. cerris*, *Q. pubescens*, *Carpinus* spp., and *F. sylvatica* were strictly similar from the point of view of fungal-order composition. Only *F. sylvatica* showed a higher level of *Russulales*. In the last four cases, the contribution of fungal orders to community organization was similar.

At the family level, the ordination of habitats overlapped with that at order level (Figure 4) despite different internal organization of the groups *C. sativa*, *Q. cerris*, *Q. pubescens*, *Carpinus* spp., and *F. sylvatica* is observed. In the grassland habitat, a peculiar fungal community structure took place, with that area dominated by Agaricaceae (18.6%), Hygrophoraceae (11.62%), Lycoperdaceae (9.30%), Bolbitiaceae (6.97%), Strophariaceae (6.97%), Hymenogasteraceae, Marasmiaceae, and Pleurotaceae (4.65% each). With shifting in forest ecosystems, family composition dramatically changes. In *A. alba*, the dominant families were Boletaceae (13.75%), Russulaceae (10%), Hydnaceae (7.5%), Amanitaceae, Gomphaceae, and *Polyporales* (5% for each taxon). Within the allochthonous conifer habitat, the most frequent group was Agaricaceae, accounting for 9.78% of the total, followed by Russulaceae (8.64%), Suillaceae (7.40%), Boletaceae and Gomphaceae (6.17%), and Hydnaceae (4.93%). Among angiosperms, Boletaceae dominated the fungal community (8.9%), followed by Amanitaceae and Strophariaceae (4.95%), Hydnaceae, Tricholomataceae and Entolomataceae, Lyophyllaceae, Mycenaceae, and Lycoperdaceae (2.97%). In *F. sylvatica*, the fungal community was organized with a high level of Russulaceae (13.79%), followed by Boletaceae (12.31%), Tricholomataceae (5.9%), and Polyporaceae and Cortinariaceae (5.4%). Other families such as Amanitaceae and Hydnaceae were present at 4.94%. In *Q. cerris*, *Q. pubescens*, *C. Sativa*, and *Carpinus* spp. environments, fungal families were similarly organized, with Boletaceae as the first taxa (14–10%), followed by Russulaceae (9–10%) and Amanitaceae and Tricholomataceae (7–5%).

Habitat similarities according to ecological categories partially mirrored those observed at family and order levels (Figure 5). Grassland remain the most distant habitat dominated by saprotrophs species. St (terricolous saprotrophs, 44.18%) was the more frequent trophic mode, followed by Sl (Saprotrophs on litter, 25.58) and Sd (Saprotrophs on dung, 23.25%). Small percentage of Sm (Saprotrophs on mosses, 2.32%) and Sw (Saprotrophs on wood, 4.65%). In forested environments, organization of the trophic modes of fungal community dramatically changed because of the presence of Ecm (Ectomycorrhizal) taxa in a range of 37–61% relative abundance. Mixed broad-leaved forests were the most dissimilar habitats among the forested environments, with Ecm at 37.62% followed by Sw and Sl (28.7%, each), St (3.96%), Pn (1.98%) and Pm (0.99%). *F. sylvatica* was clustered together with *Carpinus* spp. and *Q. pubescens.* These habitats were strongly dominated by Ecm taxa (60.39–64.00%), followed by Sw (18.22–19.8%), Sl (10.89–15.76%), St (1.47–4.00%) and Pn (Necrotroph parasites, 0.98–4.95%). Pm (parasite on mushrooms); Sd and Sm appeared only in *F. sylvatica* with percentage below 1% of the total. The same organization was observed in *Q. cerris* and *C. sativa* habitats but with lower level of ECM taxa compared to the previous group. Ecm taxa dominated among fungal trophic strategies with values between (52.5–53.0%); Sw was the second category ranging from (21.96–22.0%). Four other categories were observed within the habitats, specifically Sl, St, Pn and Pm with values ranges of 15.25–16.66%, 4.54–5.64%, 3.0–3.9%, and 0.56–0.76%, respectively. Finally, conifer dominated habitats were clustered together because of their similar dominant fungal ecological categories but with specific changes as Sc (Saprotrophs on cones) that were present only in the allochthonous conifer habitat (3.70%). Within those habitats, fungal trophic mode categories were dominated by Ecm species (46.9–48.75%), Sl (20.98–21.25%), Sw (20.00–17.28%). Other categories present in minor contribution were Pn (4.9–5.00%), St (3.75–6.17%) and Sm was present only in the *A. alba* environment (1.25%).

## 4. Discussion

The importance of fungal checklists lies in their substantial contribution to fungal ecology, systematics, and forestry sciences as providing helpful information for environmental management of conservation areas [5,18,30]. In this study, an overview of macrofungal occurrence within the Taburno-Camposauro massif is presented (Table 1), with some representative species shown in Figure 6, Figure 7 and Figure 8. This research serves as a valuable repository, detailing fungal biodiversity across diverse habitats and providing essential data for assessing the biogeographic distribution of macrofungi in the Mediterranean basin. The data presented here can inform future directions in conservation policies for different management authorities.

As macrofungi may serve as bioindicator species [7,31,32], the understanding of their presence and distribution became an essential factor to monitor the ecological state of natural environments. Future censuses of macrofungi in time series can provide valuable insights into the changes of biotic populations and communities in climate-change conditions, and can help to assess the overall health of ecosystems [10,33]. In addition to their utility in environmental conservation policies, macrofungi represent a biological reservoir of high-value food and medicinal products. Considered delicacies in many cuisines globally, they also play integral roles in local cultures and folklore [34]. For these reasons, monitoring macrofungi can support the assessment of important ecosystem services as valuable natural food resources and the efficiency of natural environmental benefits [35].

Regarding new records in Campania, 195 additional observations have been provided for the region, as noted in previous publications on the topic. The most recent checklist for the region was published by Violante in 2002 [20] by cumulating data from local checklists in the five provinces of the Campania region. However, since the initial work of Violante, several informative works have been produced, and numerous collections are preserved in private and mycological-association herbaria. Before the central review of Violante and colleagues, data on macrofungi in Campania were present in dated works by Briganti, Comes, Terracciano, Trotter, and Domenico Saccardo. It is likely that many species, here reported as first records, were in there as previously observed in the environments of Campania region. A comprehensive work covering the mycological history in Campania and the species found in it could be considered in a dedicated project.

Most of the observations in the current checklist belong to Basidiomycetes, with a smaller portion belonging to Ascomycota. The disparity in data distribution is primarily attributed to challenges in observing the latter. In numerous instances, Ascomycota produce small-sized sporophores, hardly detectable in naked-eye surveys. Furthermore, the reproductive period of these taxa, concentrated in early spring and late fall, is affected by the lower frequency of sampling due to prohibitive climatic conditions in mountain environments. To address this gap in knowledge, specific surveys should be planned in the future to thoroughly assess the presence of this important phylum.

Concerning macrofungal diversity, the site with the highest number of records was the habitat dominated by *F. sylvatica*, followed by *Q. cerris*, *C. sativa*, the mixed broadleaved forests, *Q. pubescens*, allochthonous conifers, *A. alba*, *Carpinus* spp., and grassland with the lowest value. The difference in the number of fungal taxa could be speculated to be influenced by the surface area of the habitat and its vegetation composition examined in the survey. However, the most representative habitat, in terms of dimension, is that dominated by *Carpinus* spp., which accounts for a relatively low number of fungal species compared to the other forested habitats. An observed discrepancy exists between the vegetation type data from satellite images and those observed on-site. Extensive patches of *Q. cerris*, *Q. pubescens*, and cultivated *C. sativa* were frequently present in the area designated as *Carpinus* spp. forest, leading to an overrepresentation of its presence. Concerning *F. sylvatica* and mixed broad-leaved forests, the higher number of fungal taxa could be attributed to environmental conditions favoring macrofungal reproduction. *F. sylvatica* acts as an ecosystem engineer, buffering extreme environmental conditions like drought and frost [36,37], often facilitating the development of fungal sporophores [5]. Similarly, mixed broad-leaved forests are predominantly found in riparian zones and valleys, providing optimal hygrometric conditions for the development of fungal sporophores [38]. On the contrary, stable grassland exhibited a lower number of fungal taxa despite its vast presence in the massif. Macrofungi, primarily Basidiomycetes, are slow-growing, late-stage decomposers of recalcitrant organic matter that show a preference for undisturbed conditions [39,40]. Particularly noteworthy, a high level of disturbance by boar activity and excessive grazing was observed in the grassland habitats of the massif, namely Piano Melaino, Piano Cepino (Taburno mount), and Piana of Camposauro (Camposauro mount). The consequent disruption of vegetation cover can produce unsuitable conditions for fungal growth, with the exception of perturbance-related taxa such as *Volvopluteus gliochephalus* (Pluteaceae), [41,42], recorded in grassland habitats of Taburno-Camposauro massif. The highest diversity of fungal taxa was observed in areas dominated by Fagaceae trees, with the sole exception of *Q. pubescens.* It may be thought that *Q. pubescens* habitats, positioned in foothill and xeric settings, may experience disturbance related to human activities and fires. Evidence of past fires, such as carbonized plant biomass, reduced woody necromass, and fire-associated plant species (*Spartium junceum*) support the hypothesis. However, the absence of a fungal ecological category of saprotrophs on burnt ground (Sbg) suggests that these events are not recent or frequent, potentially demonstrating the success of fire-prevention strategies by management authorities in recent years.

When analyzing macrofungal community structure, the grassland habitat clearly differed in species families compared to the forest habitat, suggesting that ecosystem peculiarities constitute a principal evolutive driver in macrofungi. Among forest habitats, a distinct difference is observed between Fagacean and *Carpinus* spp. forests and others because of a higher contribution of Amanitaceae, Hydnangiaceae, Sclerodermataceae, Hygrophoraceae, Tricholomataceae, Boletaceae, and Cortinariaceae. In coniferous trees, Agaricaceae, Tapinellaceae, Hydnaceae, Phylacriaceae, Gomphaceae, Gloeophyllaceae, and a higher contribution of *Polyporales* form the main structure of the macrofungal community at the family level. Specific associations, such as Suillaceae in allochthonous conifer forest, and Inocybaceae, Gloeophyllaceae, and Omphalotaceae with *A. alba* forests, clearly distinguish the two conifer-dominated habitats. Interestingly, mixed broad-leaved forests host a higher relative abundance of Morchellaceae, Pleurotaceae, Lyophyllaceae, Strophariaceae, Lycoperdaceae, Psathyrellaceae, and Entolomataceae. Some fungal families, including Mycenaceae, exhibit a shared pattern of association with broad-leaved species and Fagaceae habitats, while Russulaceae are more abundant between Fagaceae and coniferous trees.

The overall representation of fungal community structuration at the family level suggests a relation with ecological guilds and trophic modes of fungi within different environments. Fagacean trees and *Carpinus* spp. dominated forests are very similar regarding their taxonomic and ecological composition of macrofungal community. Amanitaceae, Hydnangiaceae, Sclerodermataceae, Hygrophoraceae, Tricholomataceae, Boletaceae, and Cortinariaceae reflect the high dependency of these tree species on ectomycorrhizal symbiosis (ECM), also as a response to plant-soil feedback phenomena regulating mono-dominanace in these environments [43,44,45]. ECM symbiosis has evolved multiple times since the Early Jurassic, occurring twice in gymnosperms and 28 times in angiosperms [46]. However, fungal taxonomic distance proves to be a poor predictor of ecological guild affiliation. For instance, *Russula* and *Tuber* genera can both form ECM symbiosis despite their distant phylogenetic relationship. The observed overlap in fungal taxa in fagacean habitats likely results from an event of symbiotic association by a common ancestor and subsequent speciation events of those trees. In contrast, for *Carpinus* spp., the close association of habitat, shared with fagacean species, led to a partial exchange of symbiotic mycobiota.

Coniferous tree species also show the presence of these taxa, as they are dependent on ECM symbiosis. However, a more specialized group of fungi (Suillaceae) was observed, mixed with many saprotrophic fungal species (Agaricaceae). An alternative explanation for the lower presence of ECM compared to fagacean-dominated habitats may be attributed to the unique conditions of coniferous-dominated habitats in the Taburno-Camposauro massif. Allochthonous conifer habitats resulted from the patchy reafforestation policies that were promoted by territorial authorities from 1970 to 1990. The two-needle pines, *P. menziesi*, and *Larix* spp., are species with different geographic origins, so morphological isolation and habitat fragmentation may limit the continuous input of spore dissemination for specialized symbiotic hosts [47]. On the other hand, *Cupressus* spp. are present within the reafforested areas but are not able to host ECM symbiosis [48], allowing for the occurrence of several Agaricacean saprotrophic species. For *A. alba*, the forest is strictly localized in the Taburno massif, with limited land area. This forest habitat is a relic of the more extensive areas planted during Bourbon rule in Southern Italy in 1846. The area, known as the State Forest of Taburno, experiences isolation from other patches of *A. alba* with a natural distribution (the nearest being Pollino massif in Calabria to the south ~182 km and Mainarde massif to the north ~73 km) and is bordered by other massifs or Apennine disjunction. This likely prevents efficient and continuous wind dispersal of specific ECM taxa. Additionally, *A. alba* forests are characterized by a high presence of old, growing individuals and plant necromass. These conditions allow the accumulation of fungal pathogens such as Phylacriaceae (which includes the genus *Armillaria*) and *Heterobasidion annosum* as well as families with known saprotrophic ability on wood residues like *Polyporales*, Tapinellaceae, and Gloeophyllaceae. In addition to diluting the relative abundance of ECM species, the higher presence of saprotrophic and pathogenic taxa suggests an imbalance in soil mycobiota. This representation may be explained by a weakening of the Gadgil effect in ECM forest environments. The Gadgil effect theory explains the relationships between ECM species and saprotrophs in the forest floor, stating that symbiotic species, when tethered to the plant host, gain a competitive advantage over litter saprotrophs [49,50]. In the case of the coniferous trees of the Taburno-Camposauro massif, the pathogen accumulation may imbalance the richness in ECM fungal species, leaving space for free-leaving fungal saprotrophs, although dedicated studies are needed to describe the phenomenon in detail.

The same explanation may be taken into consideration for mixed broad-leaved forests. In this habitat, fungal families such as Morchellaceae, Pleurotaceae, Lyophyllaceae, Strophariaceae, Lycoperdaceae, Psathyrellaceae, Entolomataceae, and Mycenaceae appear with increased frequency. Those taxa are mainly saprotrophic on various substrates, exploiting the decreased presence or activities of ECM species. Mixed broad-leaved forests encompass riparian habitats where a moderate level of disturbance can destabilize ECM fungal communities and host plant species that are not able to form the symbiosis (*Acer*, *Fraxinus*) [29,48]. These conditions allow the opening of niches for several saprobic species.

Finally, grassland habitats were considered separately due to the obvious difference resulting from the absence of trees and potential symbiotic hosts. In this condition, free-living saprotrophs dominate the soil space, mainly divided into taxa belonging to Agaricaceae, Hygrophoraceae, Lycoperdaceae, Bolbitiaceae, Strophariaceae, Marasmiaceae, Hymenogasteraceae, Pleurotaceae, and Psathyrellaceae. All these taxa have developed adaptations to the grassland environment, decomposing soil humus, hemicellulose/cellulose-based litter residuals, or exploiting dung released by grazing herbivores. Notably, the conservation state of grassland is also assessable by the presence of fairy rings (circular colonies of expanding mycelium marked by changing vegetation or sporophores) that occur in regular circular patterns only in low-perturbed conditions [31,51]. The presence of several species belonging to the *Agaricus* genus, as well as *Marasmius oreades* and *Calocybe gambosa*, was observed [13,51]; that, however, does not form a complete circular pattern but a fragmented one, indicating a perturbed state in grassland habitats.

Future research efforts should offer a more comprehensive point of view of macrofungal diversity in Taburno-Camposauro massif compared with the present checklist. Primarily due to challenges in species assessment, especially for species such as many *Cortinarius* and *Russula*, and secondly because of the limited period in which observations were collected, a period of 10–20 years could offer a more detailed view of the fungal diversity in the area. Additionally, checklists accompanied by fungal frequency data can play a crucial role in evaluating the conservation status of numerous fungal species, many of which are currently not represented in the IUCN Red List of Threatened Species (as of February 2024, numbering 781 species). These checklists can contribute significantly to assessments of fungal biogeography and diversity in the Campania region and in the counties of Mediterranean Basin.

## 5. Conclusions

In this study, the macrofungal community in the Taburno-Camposauro massif was documented, spanning eight forested and one grassland habitats. A total of 453 fungal species, introducing new records for the Campania region, were identified. However, a more comprehensive investigation, integrating historical data and disseminating studies, is required to fully understand macrofungal diversity in the area and in the Campania region. The checklist provided may serve as a valuable resource for local management authorities, offering insights to formulate specific management policies. Moreover, authorities in other regions and conservation areas can adopt this methodology to assess fungal diversity in various habitats in order to understand their ecological status.

## Figures and Tables

**Figure 2 jof-10-00275-f002:**
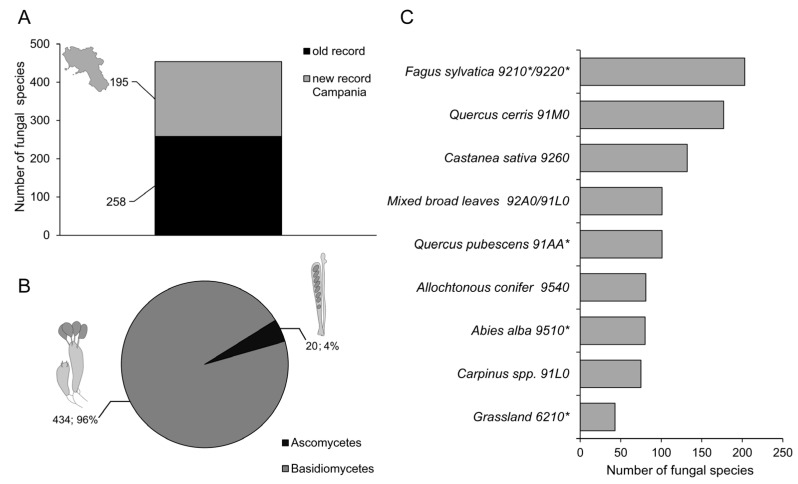
(**A**), stacked bar plot of the total number of fungal species observed in Taburno-Camposauro Regional Park in the present survey. Black bars indicate species records already collected for Campania region. Grey bars indicate new species for Campania. (**B**), relative abundance of Basidiomycetes and Ascomycetes fungal species recorded in the present survey. (**C**), number of fungal species divided according to habitats included in the survey. Asterisks in habitat codes indicate priority habitats according to natura 2000 classification.

**Figure 3 jof-10-00275-f003:**
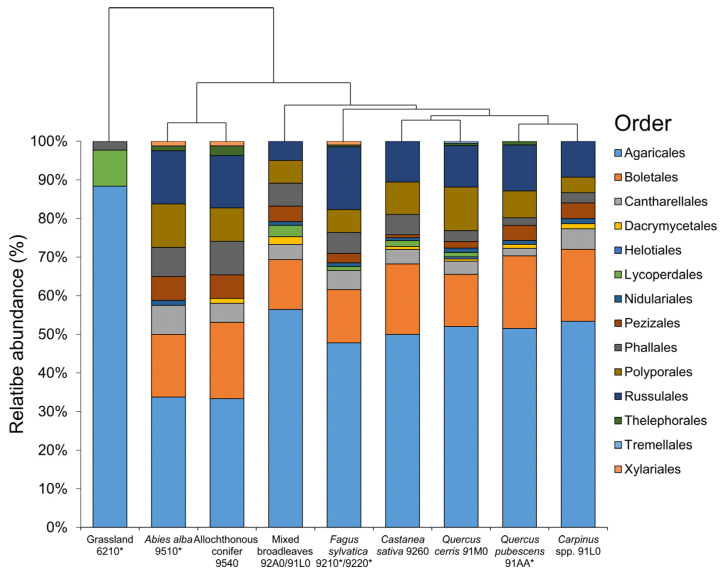
Stacked bar plot showing relative abundance of fungal orders in Taburno-Camposauro Pegional Park, divided according to habitats included in the survey. Habitats were reordered according to clustering dendrogram based on complete linkage of Bray–Curtis similarity values. Asterisksin habitat codes indicate priority habitats according to natura 2000 classification.

**Figure 4 jof-10-00275-f004:**
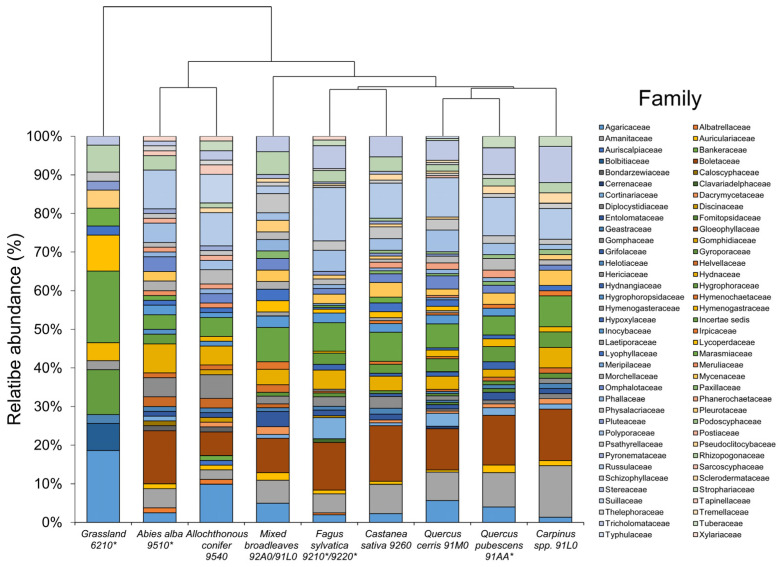
Stacked bar plot showing relative abundance of fungal families in Taburno-Camposauro Regional Park, divided according to habitats included in the survey. Habitats were reordered according to clustering dendrogram based on complete linkage of Bray–Curtis similarity values. Asterisks in habitat codes indicate priority habitats according to Natura 2000 classification.

**Figure 5 jof-10-00275-f005:**
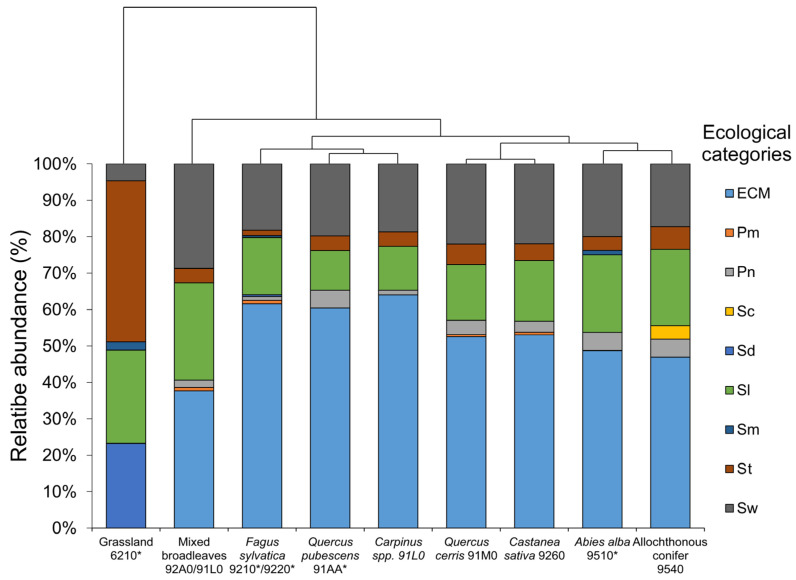
Stacked bar plot showing relative abundance of trophic mode (ecological categories) of fungal taxa in Taburno-Camposauro Regional Park, divided according to habitats included in the survey. Habitats were reordered according to clustering dendrogram based on complete linkage of Bray–Curtis similarity values. In legend, ecological categories codes represent: Ecm = Ectomycorrhizal, Pm = Parasite on mushrooms, Pn = Necrotroph parasites, Sd = Saprotrophs on dung, Sc = Saprotrophs on cones, Sl = Saprotrophs on litter, Sm = Saprotrophs on mosses, St = Terricolous saprotrophs, and Sw = Saprotrophs on wood. Asterisks in habitat codes indicate priority habitats according to Natura 2000 classification.

**Figure 6 jof-10-00275-f006:**
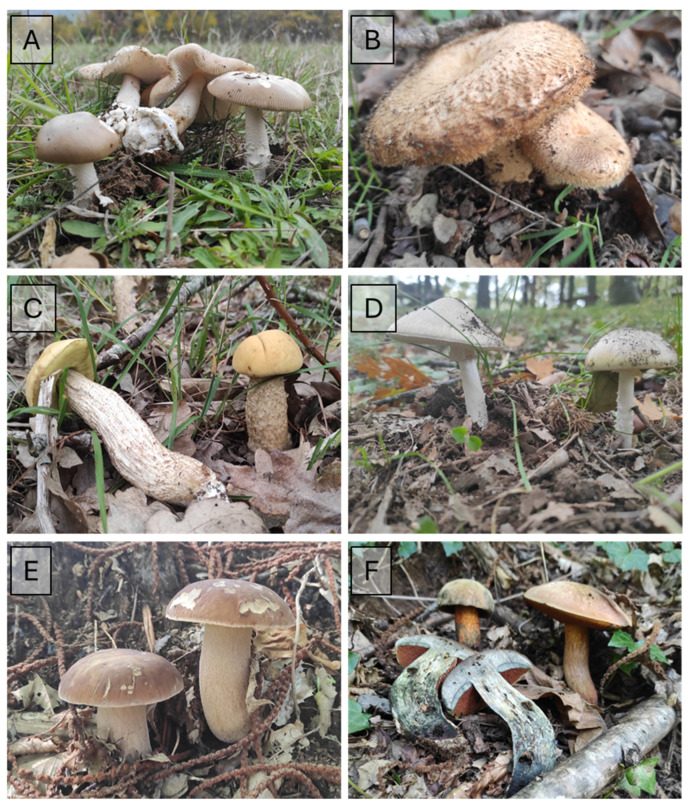
Some representative taxa in Taburno-Camposauro massif for: *Q. pubescens* habitat (**A**) *Amanita ceciliae* (Berk. & Broome) Bas. (**B**) *Lactarius mairei Malençon*; *Q. cerris habitat* (**C**) *Leccinellum crocipodium* (Letell.) Della Magg. & Trassin. (**D**) *Amanita phalloides* (Vaill. ex Fr.) Link.; *C. sativa* habitat (**E**) *Boletus reticulatus* Schaeff. (**F**) *Suillellus luridus* (Schaeff.) Murrill.

**Figure 7 jof-10-00275-f007:**
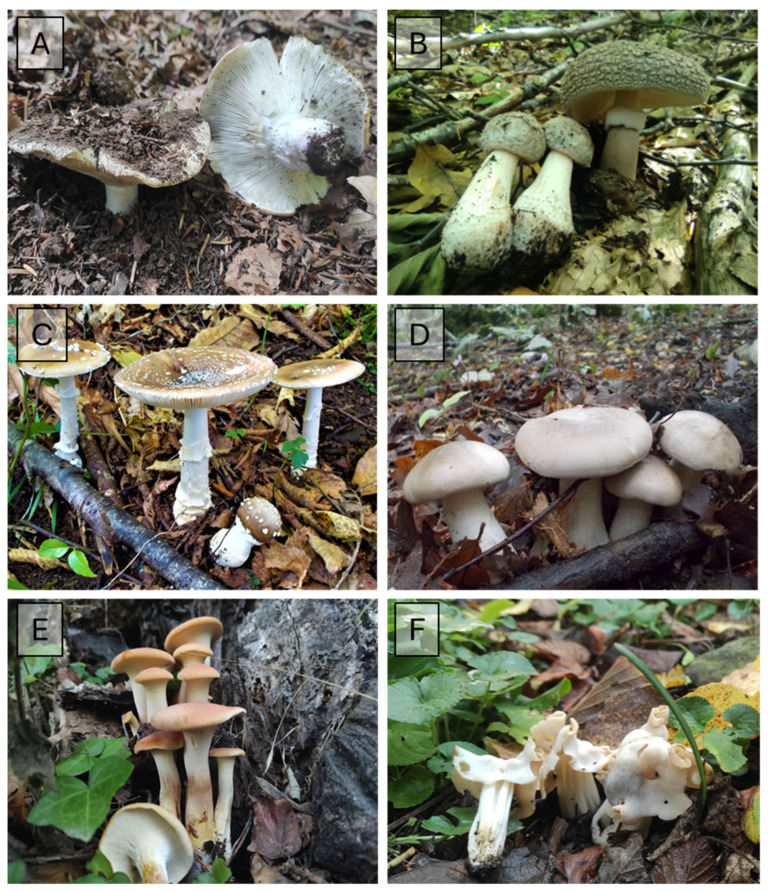
Some representative taxa in Taburno-Camposauro massif for: *F. sylvatica* habitat (**A**) *Russula delica* Fr. (**B**) *Amanita rubescens* Pers.; *Carpinus* spp. *habitat* (**C**) *Amanita pantherina* (DC.) Krombh. (**D**) *Clitocybe nebularis* (Batsch) P. Kumm.; Mixed broad-leaved habitat (**E**) *Neolentinus cyathiformis* (Schaeff.) Della Magg. & Trassin. (**F**) *Helvella crispa* (Scop.) Fr.

**Figure 8 jof-10-00275-f008:**
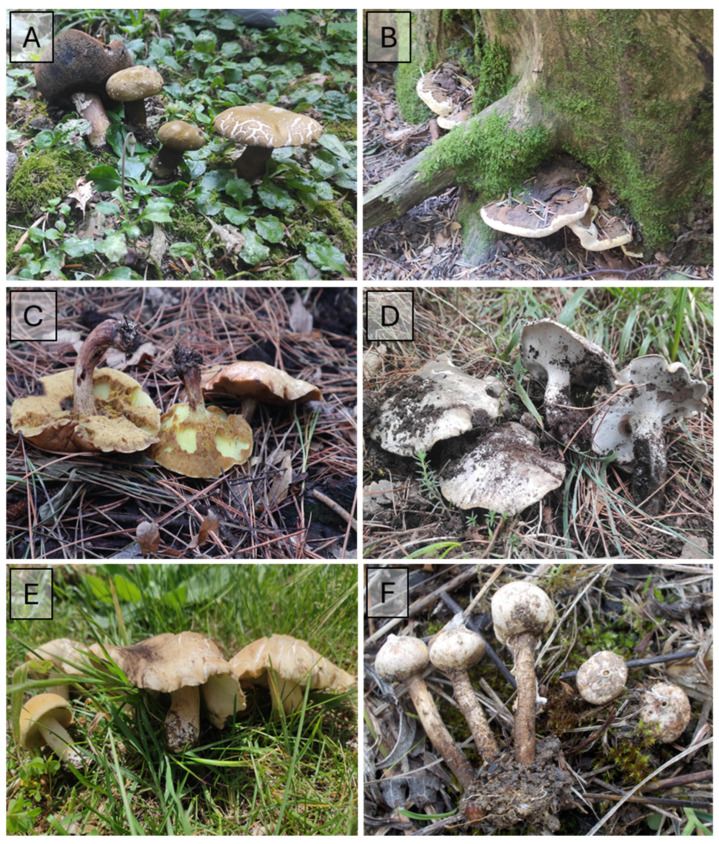
Some representative taxa in Taburno-Camposauro massif for: *A. alba* habitat (**A**) *Porphyrellus porphyrosporus* (Fr.) E.-J. Gilbert. (**B**) *Heterobasidion annosum* (Fr.) Bref.; Allochthonous conifers *habitat* (**C**) *Suillus granulatus* (L.) Roussel (**D**) *Boletopsis grisea* (Peck) Bondartsev & Singer; Grassland habitat (**E**) *Calocybe gambosa* (Fr.) Donk (**F**) *Tulostoma brumale* Pers.

**Table 1 jof-10-00275-t001:** List of recorded taxa according to order and family divided by habitats of Taburno-Camposauro massif. X indicates the presence of the taxa within a specific habitat. (Eco) represent the trophic mode (ecological categories) of the taxa divided in: Ec = Ectomycorrhizal, Pm = Parasite on mushrooms, Pn = Necrotroph parasites, Sd = Saprotrophs on dung, Sh = Saprotrophs on humus, Sl = Saprotrophs on litter, Sm = Saprotrophs on mosses, St = Terricolous saprotrophs, Sw = Saprotrophs on wood. Asterisks associated to habitat codes indicate priority habitats according to Natura 2000 classification.

Taxon	*Fagus sylvatica* 9210 */9220 *	*Abies alba* 9510 *	Allochthonous Conifers 9540	*Quercus cerris* 91M0	*Quercus pubescens* 91AA *	*Carpinus* spp. 91L0	*Castanea sativa* 9260	Mixed Broadleaves 92A0/91L0	Grassland 6210 *	Order (.ales)	Family (.aceae)	Eco
*Abortiporus biennis (Bull.) Singer*				x	x	x	x			Polypor.	Podoscyph.	Sw
*Agaricus arvensis Schaeff. **					x			x	x	Agaric.	Agaric.	St
*Agaricus augustus Fr. var. augustus **				x	x					Agaric.	Agaric.	St
*Agaricus bisporus (J. E. Lange) Imbach*									x	Agaric.	Agaric.	St
*Agaricus bresadolanus Bohus*				x		x				Agaric.	Agaric.	St
*Agaricus campestris L. **									x	Agaric.	Agaric.	St
*Agaricus moelleri Wasser **			x							Agaric.	Agaric.	St
*Agaricus sylvaticus Schaeff. **	x			x						Agaric.	Agaric.	St
*Agaricus sylvicola (Vittad.) Peck **				x						Agaric.	Agaric.	St
*Agaricus xanthodermus Genev. subsp. xanthodermus **			x							Agaric.	Agaric.	St
*Agrocybe molesta (Lasch) Singer*							x			Agaric.	Strophari.	St
*Agrocybe praecox (Pers.) Fayod **								x	x	Agaric.	Strophari.	St
*Albatrellus ovinus (Schaeff.) Kotl. & Pouzar **		x	x							Russul.	Albatrell.	Ec
*Aleuria aurantia (Pers.) Fuckel*		x					x			Peziz.	Pyronemat.	St
*Amanita caesarea (Scop.) Pers. **				x	x	x	x	x		Agaric.	Amanit.	Ec
*Amanita ceciliae (Berk. & Broome) Bas **				x	x	x				Agaric.	Amanit.	Ec
*Amanita citrina Pers. **	x	x	x	x	x	x	x	x		Agaric.	Amanit.	Ec
*Amanita crocea (Quél.) Singer **				x		x	x			Agaric.	Amanit.	Ec
*Amanita echinocephala (Vittad.) Quél. **	x			x						Agaric.	Amanit.	Ec
*Amanita eliae Quél. **				x	x	x	x			Agaric.	Amanit.	Ec
*Amanita excelsa (Fr.) Bertill. **				x						Agaric.	Amanit.	Ec
*Amanita franchetii (Boud.) Fayod*				x	x	x		x		Agaric.	Amanit.	Ec
*Amanita fulva Fr. **							x			Agaric.	Amanit.	Ec
*Amanita muscaria (L.) Lam. **	x									Agaric.	Amanit.	Ec
*Amanita pantherina (DC.) Krombh. **	x	x		x	x	x	x	x		Agaric.	Amanit.	Ec
*Amanita phalloides (Vaill. ex Fr.) Link **	x			x	x	x	x	x		Agaric.	Amanit.	Ec
*Amanita porphyria Alb. & Schwein.*	x									Agaric.	Amanit.	Ec
*Amanita rubescens Pers. **	x	x		x	x	x	x	x		Agaric.	Amanit.	Ec
*Amanita vaginata (Bull.) Lam. **	x	x		x	x	x	x			Agaric.	Amanit.	Ec
*Amanita verna Bull. ex Lam. **	x			x			x			Agaric.	Amanit.	Ec
*Amaropostia stiptica (Pers.) B.K. Cui, L.L.*		x	x							Polypor.	Posti.	Sw
*Armillaria cepistipes Velen.*				x	x					Agaric.	Physalacri.	Pn
*Armillaria gallica Marxm. & Romagn.*		x	x							Agaric.	Physalacri.	Pn
*Armillaria mellea (Vahl) P. Kumm. **	x			x	x	x	x	x		Agaric.	Physalacri.	Pn
*Arrhenia spathulata (Fr.) Redhead*	x	x								Agaric.	Hygrophor.	Sm
*Aspropaxillus candidus (Bres.) M.M. Moser*		x	x							Agaric.	Tricholomat.	St
*Aspropaxillus giganteus (Sowerby) Kühner & Maire **									x	Agaric.	Tricholomat.	St
*Asterophora lycoperdoides (Bull.) Ditmar*	x			x			x	x		Agaric.	Lyophyll.	Pm
*Astraeus hygrometricus (Pers.) Morgan*				x	x	x				Bolet.	Diplocystidi.	St
*Aureoboletus gentilis (Quél.) Pouzar **						x				Bolet.	Bolet.	Ec
*Auricularia auricula-judae (Bull.) Quél.*	x	x	x	x	x		x	x		Russul.	Auriculari.	Sw
*Auricularia mesenterica (Dicks.) Pers. **	x				x	x		x		Russul.	Auriculari.	Sw
*Auriscalpium vulgare Gray **			x							Russul.	Auriscalpi.	Sc
*Baorangia emileorum (Barbier) Vizzini, Simonini & Gelardi*							x			Bolet.	Bolet.	Ec
*Bjerkandera adusta (Willd.) P. Karst. **		x	x	x			x			Polypor.	Phanerochaet.	Pn
*Bjerkandera fumosa (Pers.) P. Karst. **				x	x		x			Polypor.	Phanerochaet.	Pn
*Bolbitius titubans (Bull.) Fr.*									x	Agaric.	Bolbiti.	Sd
*Boletopsis grisea (Peck) Bondartsev & Singer*			x							Thelephor.	Banker.	Ec
*Boletus aereus Bull. **				x	x	x	x	x		Bolet.	Bolet.	Ec
*Boletus edulis Bull. **	x									Bolet.	Bolet.	Ec
*Boletus pinophilus Pilát & Dermek **		x								Bolet.	Bolet.	Ec
*Boletus reticulatus Schaeff. **	x	x	x			x	x	x		Bolet.	Bolet.	Ec
*Bovista plumbea Pers. **									x	Lycoperd.	Lycoperd.	St
*Bovistella utriformis (Bull.) Demoulin & Rebriev*									x	Lycoperd.	Lycoperd.	St
*Butyriboletus appendiculatus (Schaeff.) D. Arora & J.L. Frank **	x			x				x		Bolet.	Bolet.	Ec
*Butyriboletus fechtneri (Velen.) D. Arora & J.L. Frank **	x									Bolet.	Bolet.	Ec
*Butyriboletus pseudoregius (Heinr. Huber) D. Arora & J.L. Frank*				x			x			Bolet.	Bolet.	Ec
*Butyriboletus regius (Krombh.) D. Arora & J.L. Frank **	x			x			x			Bolet.	Bolet.	Ec
*Byssomerulius corium (Pers.) Parmasto*				x	x	x	x			Polypor.	Irpic.	Sw
*Caloboletus calopus (Pers.) Vizzini **	x	x		x		x	x			Bolet.	Bolet.	Ec
*Caloboletus radicans (Pers.) Vizzini **	x			x			x			Bolet.	Bolet.	Ec
*Calocera cornea (Batsch) Fr. **				x	x	x	x	x		Dacrymycet.	Dacrymycet.	Sw
*Calocera viscosa (Pers.) Fr. **			x					x		Dacrymycet.	Dacrymycet.	Sw
*Calocybe gambosa (Fr.) Donk **		x	x	x			x	x	x	Agaric.	Lyophyll.	st
*Caloscypha fulgens (Pers.: Fr.) Boud.*		x								Peziz.	Caloscyph.	Sl
*Calvatia cyathiformis (Bosc) Morgan **									x	Lycoperd.	Lycoperd.	St
*Candolleomyces candolleanus (Fr.) D. Wächt. & A. Melzer*	x	x	x	x		x	x	x		Agaric.	Psathyrell.	Sw
*Cantharellus alborufescens (Malençon) Papetti & S. Alberti*	x			x	x	x	x	x		Cantharell.	Hydn.	Ec
*Cantharellus amethysteus (Quél.) Sacc.*	x	x								Cantharell.	Hydn.	Ec
*Cantharellus cibarius Fr. var. cibarius **	x	x		x			x	x		Cantharell.	Hydn.	Ec
*Cantharellus cinereus (Pers.) Fr.*	x			x						Cantharell.	Hydn.	Ec
*Cantharellus ferruginascens P.D. Orton*	x						x			Cantharell.	Hydn.	Ec
*Cantharellus friesii Quél. **	x			x						Cantharell.	Hydn.	Ec
*Cantharellus pallens Pilát*	x	x	x	x	x	x	x	x		Cantharell.	Hydn.	Ec
*Cerioporus meridionalis (A. David) Zmitr. & Kovalenko*	x			x						Polypor.	Polypor.	Sw
*Cerioporus squamosus (Huds.) Quél.*	x			x			x			Polypor.	Polypor.	Sw
*Cerrena unicolor (Bull.) Murrill*				x						Polypor.	Cerren.	Sw
*Chalciporus piperatus (Bull.) Bataille **	x									Bolet.	Bolet.	Ec
*Chroogomphus rutilus (Schaeff.) O.K. Mill. **			x							Bolet.	Gomphidi.	Ec
*Clathrus ruber P. Micheli ex Pers. **		x	x	x	x			x		Phall.	Phall.	Sl
*Clavariadelphus pistillaris (L.) Donk*	x									Phall.	Clavariadelph.	Ec
*Clavariadelphus truncatus Donk **	x									Phall.	Clavariadelph.	Ec
*Clavulina coralloides (L.) J. Schröt. **	x					x				Cantharell.	Hydn.	Sw
*Clavulina rugosa (Bull.) J. Schröt.*		x								Cantharell.	Hydn.	Sl
*Clitocybe costata Kühner & Romagn. **	x						x			Agaric.	Incertae sedis	Sl
*Clitocybe dealbata (Sowerby) P. Kumm. **				x			x			Agaric.	Incertae sedis	Sl
*Clitocybe infundibuliformis (Schaeff.) Quél.*				x						Agaric.	Incertae sedis	Sl
*Clitocybe metachroa (Fr.) P. Kumm. **			x					x		Agaric.	Incertae sedis	Sl
*Clitocybe nebularis (Batsch) P. Kumm. **	x	x		x	x	x		x		Agaric.	Incertae sedis	Sl
*Clitocybe odora (Bull.) P. Kumm. **	x						x			Agaric.	Incertae sedis	Sl
*Clitocybe phaeophthalma (Pers.) Kuyper*	x									Agaric.	Incertae sedis	Sl
*Clitocybe phyllophila (Pers.) P. Kumm. **			x				x			Agaric.	Incertae sedis	Sl
*Clitocybe rivulosa (Pers.) P. Kumm.*							x			Agaric.	Incertae sedis	Sl
*Clitopilus geminus (Paulet)Noordel. & Co-David*		x	x							Agaric.	Entolomat.	Ec
*Clitopilus prunulus (Scop.) P. Kumm. **	x			x	x		x	x		Agaric.	Entolomat.	Ec
*Collybia butyracea (Bull.) P. Kumm. **	x	x	x	x			x	x		Agaric.	Omphalot.	Sl
*Conocybe aporos Kits van Wav.*									x	Agaric.	Bolbiti.	Sd
*Conocybe tenera (Schaeff.) Fayod*									x	Agaric.	Bolbiti.	Sd
*Coprinellus disseminatus (Pers.) J.E. Lange **	x			x			x	x		Agaric.	Psathyrell.	Sw
*Coprinellus domesticus (Bolton) Vilgalys, Hopple & Jacq. Johnson*								x		Agaric.	Psathyrell.	Sw
*Coprinellus micaceus (Bull.) Vilgalys, Hopple & Jacq. Johnson **	x						x	x		Agaric.	Psathyrell.	Sw
*Coprinellus radians (Desm.) Vilgalys, Hopple & Jacq. Johnson **				x						Agaric.	Psathyrell.	Sw
*Coprinellus xanthothrix (Romagn.) Vilgalys, Hopple & Jacq. Johnson*				x	x					Agaric.	Psathyrell.	Sw
*Coprinopsis atramentaria (Bull.) Redhead, Vilgalys & Moncalvo **	x			x			x	x		Agaric.	Psathyrell.	Sw
*Coprinopsis picacea (Bull.) Redhead, Vilgalys & Moncalvo **	x				x					Agaric.	Psathyrell.	Sl
*Coprinus comatus (O.F. Müll.) Pers. **	x			x			x		x	Agaric.	Agaric.	Sl
*Cortinarius anserinus (Velen.) Rob. Henry **	x									Agaric.	Cortinari.	Ec
*Cortinarius atrocaeruleus M.M. Moser **	x			x	x					Agaric.	Cortinari.	Ec
*Cortinarius balteatocumatilis Rob. Henry*	x			x						Agaric.	Cortinari.	Ec
*Cortinarius camphoratus (Fr.) Fr.*	x									Agaric.	Cortinari.	Ec
*Cortinarius cinnabarinus Fr.*	x									Agaric.	Cortinari.	Ec
*Cortinarius cotoneus Fr. **	x									Agaric.	Cortinari.	Ec
*Cortinarius elegantior (Fr.) Fr. **				x						Agaric.	Cortinari.	Ec
*Cortinarius elegantissimus Rob. Henry*	x									Agaric.	Cortinari.	Ec
*Cortinarius infractus (Pers.) Fr.*	x			x			x			Agaric.	Cortinari.	Ec
*Cortinarius rapaceus Fr.*	x									Agaric.	Cortinari.	Ec
*Cortinarius splendens Rob. Henry **	x			x				x		Agaric.	Cortinari.	Ec
*Cortinarius varius (Schaeff.) Fr. **		x								Agaric.	Cortinari.	Ec
*Craterellus cornucopioides (L.) Pers. **	x			x						Cantharell.	Hydn.	Ec
*Craterellus lutescens (Fr.) Fr.*			x							Cantharell.	Hydn.	Ec
*Craterellus tubaeformis (Fr.) Quél.*		x	x							Cantharell.	Hydn.	Ec
*Crucibulum laeve (Huds.) Kambly*	x	x					x	x		Nidulari.	Incertae sedis	Sw
*Cuphophyllus pratensis (Pers.) Bon*									x	Agaric.	Hygrophor.	Sl
*Cuphophyllus virgineus (Wulfen) Kovalenko*									x	Agaric.	Hygrophor.	Sl
*Cupreoboletus poikilochromus (Pöder, Cetto & Zuccher.) Simonini, Gelardi & Vizzini*					x					Bolet.	Bolet.	Ec
*Cyanoboletus pulverulentus (Opat.) Gelardi, Vizzini & Simonini **	x			x		x	x			Bolet.	Bolet.	Ec
*Cyathus olla (Batsch) Pers. **	x			x						Nidulari.	Incertae sedis	Sl
*Cyathus striatus (Huds.) Willd. **				x	x	x				Nidulari.	Incertae sedis	Sw
*Cyclocybe cylindracea (DC.) Vizzini & Angelini **								x		Agaric.	Strophari.	Sw
*Cystoderma amianthinum (Scop.) Fayod*				x	x	x		x		Agaric.	Incertae sedis	Sl
*Cystoderma carcharias (Pers.) Fayod*	x									Agaric.	Incertae sedis	Sl
*Daedalea quercina (L.) Pers. **				x	x					Agaric.	Fomitopsid.	Sw
*Daedaleopsis confragosa (Bolton) J. Schröt. **				x			x			Polypor.	Polypor.	Sw
*Daedaleopsis nitida (Durieu & Mont.) Zmitr. & Malysheva*				x						Polypor.	Polypor.	Sw
*Daldinia concentrica (Bolton) Ces. & De Not. **				x	x					Bolet.	Hypoxyl.	Sw
*Deconica coprophila (Bull.) P. Karst.*									x	Agaric.	Strophari.	Sd
*Desarmillaria tabescens (Scop.) R.A.Koch & Aime **				x	x					Agaric.	Physalacri.	Pn
*Echinoderma asperum (Pers.) Bon*				x	x			x		Agaric.	Agaric.	Sl
*Echinoderma echinaceum (J.E. Lange) Bon*								x		Agaric.	Agaric.	Sl
*Entoloma aprile (Britzelm.) Sacc. **								x		Agaric.	Entolomat.	Ec
*Entoloma clypeatum (L.) P. Kumm. **								x		Agaric.	Entolomat.	Ec
*Entoloma lividoalbum (Kühner &Romagn.) Kubicka **	x									Agaric.	Entolomat.	Ec
*Entoloma sinuatum (Bull.) P. Kumm. **	x			x	x	x	x	x		Agaric.	Entolomat.	Ec
*Fistulina hepatica (Schaeff.) With. **							x			Agaric.	Incertae sedis	Sw
*Fomes fomentarius (L.) Fr. **	x									Polypor.	Polypor.	Pn
*Fomitopsis pinicola (Sw.) P. Karst.*		x	x							Polypor.	Polypor.	Sw
*Fuscoporia torulosa (Pers.) T. Wagner & M. Fisch.*				x			x			Polypor.	Hymenochaet.	Pn
*Galerina marginata (Batsch) Kühner*					x					Agaric.	Hymenogastr.	Sw
*Ganoderma applanatum (Pers.) Pat. **		x	x							Polypor.	Polypor.	Pn
*Ganoderma lucidum (Curtis) P. Karst. **				x	x					Polypor.	Polypor.	Pn
*Geastrum campestre Morgan*									x	Phall.	Geastr.	Sl
*Geastrum fimbriatum Fr. **	x						x			Phall.	Geastr.	Sl
*Geastrum triplex Jungh. **	x	x	x	x	x	x	x	x		Phall.	Geastr.	Sl
*Geopora sumneriana (Cooke ex W. Phillips) M. Torre*			x							Peziz.	Pyronemat.	Sl
*Gliophorus psittacinus (Schaeff.) Herink*							x			Agaric.	Hygrophor.	Sl
*Gloeophyllum abietinum (Bull.) P. Karst.*		x	x							Polypor.	Gloeophyll.	Sw
*Gloeophyllum sepiarium (Wulfen) P. Karst. **		x	x							Polypor.	Gloeophyll.	Sw
*Grifola frondosa (Dicks.) Gray **							x			Polypor.	Grifol.	Sw
*Gymnopilus junonius (Fr.) P.D. Orton **			x							Agaric.	Hymenogastr.	Sw
*Gymnopus dryophilus (Bull.) Murrill*	x			x						Agaric.	Omphalot.	Sw
*Gymnopus foetidus (Sowerby) P.M. Kirk **				x	x	x				Agaric.	Omphalot.	Sw
*Gymnopus fusipes (Bull.) Gray **				x			x	x		Agaric.	Omphalot.	Sw
*Gyromitra esculenta (Pers.) Fr. **			x							Peziz.	Discin.	Sl
*Gyromitra gigas (Krombh.) Cooke **	x									Peziz.	Discin.	Sl
*Gyroporus castaneus (Bull.) Quél. **	x			x	x	x	x	x		Bolet.	Gyropor.	Ec
*Gyroporus cyanescens (Bull.) Quél. **	x									Bolet.	Gyropor.	Ec
*Hapalopilus rutilans (Pers.) Murrill*				x	x					Polypor.	Phanerochaet.	Sw
*Hebeloma crustuliniforme (Bull.) Quél. **				x	x	x				Agaric.	Hymenogastr.	Ec
*Hebeloma sacchariolens Quél.*				x						Agaric.	Hymenogastr.	Ec
*Hebeloma sinapizans (Paulet) Gillet **	x			x						Agaric.	Hymenogastr.	Ec
*Helvella crispa (Scop.) Fr. **	x	x	x		x	x		x		Peziz.	Helvell.	Sl
*Helvella elastica Bull. **				x				x		Peziz.	Helvell.	Sl
*Hemileccinum depilatum (Redeuilh) Šutara*				x		x				Bolet.	Bolet.	Ec
*Hemileccinum impolitum (Fr.) Šutara **	x						x			Bolet.	Bolet.	Ec
*Hericium erinaceus (Bull.) Pers.*	x									Russul.	Herici.	Sw
*Heterobasidion annosum (Fr.) Bref. **		x	x							Russul.	Bondarzewi.	Pn
*Hohenbuehelia petaloides (Bull.) Schulzer*	x						x	x		Agaric.	Pleurot.	Sw
*Hortiboletus engelii (Hlavá?cek)Biketova & Wasser*								x		Bolet.	Bolet.	Ec
*Hydnum repandum L. **	x	x	x			x	x	x		Cantharell.	Hydn.	Ec
*Hygrocybe acutoconica (Clem.) Singer*									x	Agaric.	Hygrophor.	St
*Hygrocybe coccinea (Schaeff.) P. Kumm.*									x	Agaric.	Hygrophor.	St
*Hygrocybe conica (Schaeff.) P. Kumm.*									x	Agaric.	Hygrophor.	St
*Hygrocybe nigrescens (Quél.) Kühner*							x			Agaric.	Hygrophor.	St
*Hygrophoropsis aurantiaca (Wulfen) Maire*		x	x							Bolet.	Hygrophoropsid.	Ec
*Hygrophorus chrysodon (Batsch) Fr. **				x	x	x	x			Agaric.	Hygrophor.	Ec
*Hygrophorus cossus (Sowerby) Fr.*				x	x					Agaric.	Hygrophor.	Ec
*Hygrophorus discoxanthus (Fr.) Rea*	x									Agaric.	Hygrophor.	Ec
*Hygrophorus eburneus (Bull.) Fr. **	x									Agaric.	Hygrophor.	Ec
*Hygrophorus erubescens (Fr.) Fr.*				x		x				Agaric.	Hygrophor.	Ec
*Hygrophorus marzuolus (Fr.) Bres. **	x	x								Agaric.	Hygrophor.	Ec
*Hygrophorus nemoreus (Pers.) Fr.*	x									Agaric.	Hygrophor.	Ec
*Hygrophorus penarioides Jacobson & E. Larss.*				x						Agaric.	Hygrophor.	Ec
*Hygrophorus penarius Fr.*	x									Agaric.	Hygrophor.	Ec
*Hygrophorus persoonii Arnolds **				x	x	x				Agaric.	Hygrophor.	Ec
*Hygrophorus russula (Schaeff. ex Fr.) Kauffman **				x	x					Agaric.	Hygrophor.	Ec
*Hymenopellis radicata (Relhan)R.H. Petersen **	x									Agaric.	Physalacri.	Sw
*Hymenoscyphus calyculus (Fr.) W. Phillips **				x						Heloti.	Heloti.	Sw
*Hypholoma fasciculare (Huds.) P. Kumm. **	x	x		x	x	x	x	x		Agaric.	Strophari.	Sw
*Hypholoma lateritium (Schaeff.) P. Kumm.*	x	x		x	x	x	x	x		Agaric.	Strophari.	Sw
*Hypsizygus ulmarius (Bull.) Redhead*								x		Agaric.	Lyophyll.	Sw
*Imperator luteocupreus (Bertéa & Estadès) Assyov, Bellanger* et al.	x									Bolet.	Bolet.	Ec
*Imperator rhodopurpureus (Smotl.) Assyov, Bellanger, Bertéa* et al. ***				x	x	x	x			Bolet.	Bolet.	Ec
*Infundibulicybe geotropa (Bull.) Harmaja **	x			x				x		Agaric.	Incertae sedis	St
*Infundibulicybe gibba (Pers.) Harmaja **	x			x	x	x	x	x		Agaric.	Incertae sedis	Sl
*Infundibulicybe meridionalis (Bon) Pérez-De-Greg.*	x			x	x	x	x	x		Agaric.	Incertae sedis	Sl
*Inocybe amethystina Kuyper*	x							x		Agaric.	Inocyb.	Ec
*Inocybe assimilata Britzelm. **		x								Agaric.	Inocyb.	Ec
*Inocybe asterospora Quél. **	x			x						Agaric.	Inocyb.	Ec
*Inocybe geophylla (Sowerby) P. Kumm. **							x	x		Agaric.	Inocyb.	Ec
*Inocybe lanuginosa (Bull.) Kalchbr.*	x									Agaric.	Inocyb.	Ec
*Inocybe pyriodora (Pers.) P. Kumm.*				x			x			Agaric.	Inocyb.	Ec
*Inocybe whitei (Berk. & Broome) Sacc. **					x					Agaric.	Inocyb.	Ec
*Inonotus hispidus (Bull.) P. Karst. **								x		Polypor.	Hymenochaet.	Pn
*Inosperma bongardii (Weinm.) Matheny &Esteve-Rav.*				x						Agaric.	Inocyb.	Ec
*Laccaria amethystina Cooke **	x			x	x		x			Agaric.	Hydnangi.	Ec
*Laccaria bicolor (Maire) P.D. Orton **	x									Agaric.	Hydnangi.	Ec
*Laccaria laccata (Scop.) Cooke **	x			x	x					Agaric.	Hydnangi.	Ec
*Lacrymaria lacrymabunda (Bull.) Pat. **	x	x	x	x	x		x	x		Agaric.	Inocyb.	Sl
*Lactarius acerrimus Britzelm. **					x					Russul.	Russul.	Ec
*Lactarius atlanticus Bon*					x					Russul.	Russul.	Ec
*Lactarius blennius (Fr.) Fr. **	x									Russul.	Russul.	Ec
*Lactarius chrysorrheus Fr. **				x	x					Russul.	Russul.	Ec
*Lactarius deliciosus (L.) Gray **			x							Russul.	Russul.	Ec
*Lactarius fuliginosus (Fr.) Fr. **	x									Russul.	Russul.	Ec
*Lactarius mairei Malençon*				x						Russul.	Russul.	Ec
*Lactarius salmonicolor R. Heim & Leclair **		x								Russul.	Russul.	Ec
*Lactarius sanguifluus (Paulet) Fr. **			x							Russul.	Russul.	Ec
*Lactarius semisanguifluus R. Heim & Leclair **			x							Russul.	Russul.	Ec
*Lactarius subumbonatus Lindgr.*				x						Russul.	Russul.	Ec
*Lactarius zonarius (Bull.) Fr. **				x	x	x	x			Russul.	Russul.	Ec
*Lactifluus bertillonii (Neuhoff ex Z. Schaef.) Verbeken*				x			x			Russul.	Russul.	Ec
*Lactifluus piperatus (L.) Roussel **	x									Russul.	Russul.	Ec
*Lactifluus vellereus (Fr.) Kuntze **	x	x		x						Russul.	Russul.	Ec
*Lactifluus volemus (Fr.) Kuntze **				x		x	x			Russul.	Russul.	Ec
*Laetiporus sulphureus (Bull.) Murrill **				x			x	x		Polypor.	Laetipor.	Sw
*Lanmaoa fragrans (Vittad.) Vizzini, Gelardi & Simonini **							x			Bolet.	Bolet.	Ec
*Leccinellum crocipodium (Letell.) Della Magg. & Trassin. **				x						Bolet.	Bolet.	Ec
*Leccinellum pseudoscabrum (Kallenb.) Mikšík **								x		Bolet.	Bolet.	Ec
*Leccinum aurantiacum (Bull.) Gray **	x						x			Bolet.	Bolet.	Ec
*Leccinum duriusculum (Schulzer exKalchbr.) Singer*								x		Bolet.	Bolet.	Ec
*Leccinum versipelle (Fr. & Hök) Snell*								x		Bolet.	Bolet.	Ec
*Lentinus arcularius (Batsch) Zmitr.*	x			x			x			Polypor.	Polypor.	Sw
*Lentinus tigrinus (Bull.) Fr.*								x		Polypor.	Polypor.	Sw
*Lepiota clypeolaria (Bull.) P. Kumm. **	x		x	x			x			Agaric.	Agaric.	Sl
*Lepiota cristata (Bolton) P. Kumm. **			x	x						Agaric.	Agaric.	Sl
*Lepiota helveola Bres. **			x							Agaric.	Agaric.	Sl
*Lepiota ignivolvata Bousset & Joss. ex Joss.*		x	x							Agaric.	Agaric.	Sl
*Lepiota lilacea Bres.*							x	x		Agaric.	Agaric.	Sl
*Lepista irina (Fr.) H.E. Bigelow **									x	Agaric.	Incertae sedis	Sl
*Lepista nuda (Bull.) Cooke **	x			x		x		x		Agaric.	Incertae sedis	Sl
*Lepista panaeolus (Fr.) P. Karst.*									x	Agaric.	Incertae sedis	Sl
*Lepista sordida (Schumach.) Singer*	x		x					x		Agaric.	Incertae sedis	Sl
*Leratiomyces ceres (Cooke & Massee) Spooner & Bridge*				x				x		Agaric.	Strophari.	Sl
*Leucoagaricus americanus (Peck) Vellinga*									x	Agaric.	Agaric.	St
*Leucoagaricus leucothites (Vittad.) Wasser*			x						x	Agaric.	Agaric.	St
*Leucocortinarius bulbiger (Alb.& Schwein.) Singer*							x			Agaric.	Incertae sedis	Ec
*Leucopaxillus gentianeus (Quél.) Kotl.*				x						Agaric.	Tricholomat.	Sl
*Limacellopsis guttata (Pers.) Zhu L. Yang, Q. Cai & Y.Y. Cui*	x		x							Agaric.	Amanit.	Sl
*Lycoperdon echinatum Pers.*				x			x	x		Lycoperd.	Lycoperd.	Sl
*Lycoperdon excipuliforme (Scop.) Pers.*								x		Lycoperd.	Lycoperd.	Sl
*Lycoperdon mammiforme Pers.*							x	x		Lycoperd.	Lycoperd.	Sl
*Lycoperdon perlatum Pers. **	x			x						Lycoperd.	Lycoperd.	St
*Lycoperdon pratense Pers.*									x	Lycoperd.	Lycoperd.	St
*Lycoperdon pyriforme Schaeff. **	x									Lycoperd.	Lycoperd.	Sw
*Lyophyllum decastes (Fr.) Singer **				x		x				Agaric.	Lyophyll.	St
*Lyophyllum infumatum (Bres.) Kühner*							x			Agaric.	Lyophyll.	St
*Macrolepiota excoriata (Schaeff.) Wasser*									x	Agaric.	Agaric.	Sl
*Macrolepiota mastoidea (Fr.) Singer*				x	x					Agaric.	Agaric.	Sl
*Macrolepiota procera (Scop.) Singer **	x	x	x	x				x		Agaric.	Agaric.	Sl
*Marasmiellus candidus (Fr.) Singer*				x						Agaric.	Omphalot.	Sw
*Marasmius collinus (Scop.) Singer **									x	Agaric.	Marasmi.	St
*Marasmius oreades (Bolton) Fr. **									x	Agaric.	Marasmi.	St
*Marasmius rotula (Scop.) Fr. **							x			Agaric.	Marasmi.	Sw
*Marasmius wynneae Berk. & Broome **	x						x			Agaric.	Marasmi.	Sw
*Megacollybia platyphylla (Pers.) Kotl. & Pouzar **	x									Agaric.	Incertae sedis	Sl
*Melanoleuca brevipes (Bull.) Pat. **									x	Agaric.	Incertae sedis	Sl
*Melanoleuca cognata (Fr.) Konrad & Maubl.*				x						Agaric.	Incertae sedis	Sl
*Melanoleuca grammopodia (Bull.) Murrill **									x	Agaric.	Incertae sedis	Sl
*Melanoleuca melaleuca (Pers.) Murrill*									x	Agaric.	Incertae sedis	Sl
*Meripilus giganteus (Pers.) P. Karst. **	x			x						Polypor.	Meripil.	Sw
*Morchella costata Pers. **		x								Peziz.	Morchell.	Sl
*Morchella deliciosa Fr. **								x		Peziz.	Morchell.	Sl
*Morchella esculenta (L.) Pers. **		x								Peziz.	Morchell.	Sl
*Mucidula mucida (Schrad.) Pat.*	x									Agaric.	Physalacri.	Sw
*Mutinus caninus (Huds.) Fr. **	x			x				x		Phall.	Phall.	Sl
*Mycena arcangeliana Bres.*					x					Agaric.	Mycen.	Sw
*Mycena epipterygia (Scop.) Gray **				x						Agaric.	Mycen.	St
*Mycena haematopus (Pers.) P. Kumm.*	x				x	x	x			Agaric.	Mycen.	Sw
*Mycena meliigena (Berk. & Cooke) Sacc. **					x		x			Agaric.	Mycen.	St
*Mycena pelianthina (Fr.) Quél. **	x									Agaric.	Mycen.	Sl
*Mycena pura (Pers.) P. Kumm. **	x	x		x		x	x	x		Agaric.	Mycen.	Sl
*Mycena renati Quél. **	x						x	x		Agaric.	Mycen.	Sw
*Mycena rosea Gramberg **	x	x		x		x	x	x		Agaric.	Mycen.	Sl
*Mycetinis alliaceus (Jacq.) Earle ex A.W. Wilson & Desjardin **	x									Agaric.	Omphalot.	Sl
*Mycetinis scorodonius (Fr.) A.W. Wilson & Desjardin **		x								Agaric.	Marasmi.	Sl
*Neoboletus praestigiator (R. Schulz) Svetash., Gelardi, Simonini & Vizzini **	x	x	x	x						Bolet.	Bolet.	Ec
*Neoboletus xanthopus (Klofac & A. Urb.) Klofac & A. Urb.*					x					Bolet.	Bolet.	Ec
*Neofavolus alveolaris (DC.) Sotome & T. Hatt*	x									Polypor.	Polypor.	Sw
*Neolentinus cyathiformis (Schaeff.) DellaMagg. & Trassin.*								x		Polypor.	Gloeophyll.	Sw
*Omphalotus olearius (DC.) Singer **				x	x		x	x		Agaric.	Omphalot.	Sw
*Panaeolus acuminatus (P. Kumm.) Quél. **									x	Agaric.	Incertae sedis	Sd
*Panaeolus papilionaceus (Bull.) Quél. **	x								x	Agaric.	Incertae sedis	Sd
*Panaeolus semiovatus (Sowerby) S. Lundell & Nannf.*									x	Agaric.	Incertae sedis	Sd
*Paragymnopus perforans (Hoffm.) J.S. Oliveira **		x								Agaric.	Omphalot.	Sw
*Paralepista flaccida (Sowerby) Vizzini **	x									Agaric.	Incertae sedis	Sl
*Parasola plicatilis (Curtis) Redhead, Vilgalys & Hopple*									x	Agaric.	Psathyrell.	Sl
*Paxillus involutus (Batsch) Fr. **				x	x		x	x		Bolet.	Paxill.	Ec
*Paxillus rubicundulus P.D. Orton*								x		bolet.	Paxill.	Ec
*Phallus impudicus L. **	x						x	x		Phall.	Phall.	Sl
*Phellinus pomaceus (Pers.) Maire*								x		Polypor.	Hymenochaet.	Sw
*Phlebia rufa (Pers.: Fr.) M.P. Christ.*		x	x	x						Polypor.	Meruli.	Sw
*Pholiota adiposa (Batsch) P. Kumm.*	x									Agaric.	Strophari.	Sw
*Pholiota gummosa (Lasch) Singer*								x		Agaric.	Strophari.	Sw
*Pholiota squarrosa (Vahl) P. Kumm.*	x		x							Agaric.	Strophari.	Sw
*Phylloporus pelletieri (Lév.) Quél.*		x								Bolet.	Bolet.	Ec
*Picipes badius (Persoon) Zmitr. & Kovalenko*	x			x						Polypor.	Polypor.	Sw
*Picipes melanopus (Pers.) Zmitr. & Kovalenko*	x									Polypor.	Polypor.	Sw
*Pisolithus arhizus (Scop.) Rauschert*			x		x					Bolet.	Sclerodermat.	Ec
*Pleurotus cornucopiae (Paulet) Rolland **								x		Agaric.	Pleurot.	Sw
*Pleurotus eryngii (DC.) Quél. var. eryngii **									x	Agaric.	Pleurot.	Sw
*Pleurotus eryngii (DC.) Quél. var. ferulae (Lanzi) Sacc. **									x	Agaric.	Pleurot.	Sw
*Pleurotus ostreatus (Jacq.) P. Kumm. **	x					x		x		Agaric.	Pleurot.	Sw
*Pluteus cervinus (Schaeff.) P. Kumm. **				x			x			Agaric.	Plute.	Sw
*Polyporus tuberaster (Jacq. ex Pers.) Fr. **	x			x						Polypor.	Polypor.	Sw
*Porphyriellus porphyrosporus (Fr. & Hok) A.H. Sm. & Thiers*		x	x							Bolet.	Bolet.	Ec
*Porpoloma macrorhizum (Quél.) Bon Pseudoclitocybaceae*				x			x			Agaric.	Pseudoclitocyb.	Sw
*Protostropharia semiglobata (Batsch) Redhead, Moncalvo & Vilgalys*									x	Agaric.	Strophari.	Sd
*Pseudosperma rimosum (Bull.) Matheny & Esteve-Rav. **	x									Agaric.	Inocyb.	Ec
*Psilocybe coronilla (Bull.) Noordel. **									x	Agaric.	Hymenogaster.	st
*Psilocybe semilanceata (Fr.) P. Kumm.*									x	Agaric.	Hymenogastr.	Sd
*Psilocybe serbica M.M. Moser & E. Horak*									x	Agaric.	Hymenogastr.	Sd
*Ramaria aurea (Schaeff.) Quél.*	x						x			Phall.	Gomph.	Ec
*Ramaria botrytis (Pers.) Bourdot **	x	x		x			x	x		Phall.	Gomph.	Ec
*Ramaria flava (Schaeff.) Quél. **	x	x	x				x			Phall.	Gomph.	Ec
*Ramaria formosa (Pers.) Quél. **	x	x	x							Phall.	Gomph.	Ec
*Ramaria gracilis (Pers.) Quél. **			x							Phall.	Gomph.	Ec
*Ramaria pallida (Schaeff.) Ricken **			x							Phall.	Gomph.	Ec
*Ramaria stricta (Pers.) Quél. **	x	x	x	x		x	x	x		Phall.	Gomph.	Ec
*Rheubarbariboletus armeniacus (Quél.) Vizzini, Simonini & Gelardi **				x	x					Bolet.	Bolet.	Ec
*Rheubarbariboletus persicolor (H. Engel, Klofac, H. Grünert & R. Grünert) Vizzini, Simonini & Gelardi*				x	x					Bolet.	Bolet.	Ec
*Rhizopogon roseolus (Corda) Th. Fr.*							x			Bolet.	Rhizopogon.	Ec
*Rhodocollybia maculata (Alb.& Schwein.) Singer **		x	x							Agaric.	Omphalot.	Sl
*Rhodotus palmatus (Bull.) Maire*								x		Agaric.	Physalacri.	Sw
*Rubroboletus legaliae (Pilát & Dermek) DellaMagg. & Trassin. **	x						x			Bolet.	Bolet.	Ec
*Rubroboletus lupinus (Fr.) Costanzo, Gelardi, Simonini & Vizzini **					x		x			Bolet.	Bolet.	Ec
*Rubroboletus pulchrotinctus (Alessio) Kuan Zhao & Zhu L. Yang*							x			Bolet.	Bolet.	Ec
*Rubroboletus rhodoxanthus (Krombh.) KuanZhao & Zhu L. Yang*	x			x	x					Bolet.	Bolet.	Ec
*Rubroboletus rubrosanguineus (Cheype) Kuan Zhao & Zhu L. Yang*	x	x								Bolet.	Bolet.	Ec
*Rubroboletus satanas (Lenz) Kuan Zhao &Zhu L. Yang **	x			x	x		x			Bolet.	Bolet.	Ec
*Russula acrifolia Romagn. **	x									Russul.	Russul.	Ec
*Russula adusta (Pers.) Fr.*		x								Russul.	Russul.	Ec
*Russula albonigra (Krombh.) Fr. **				x	x	x				Russul.	Russul.	Ec
*Russula alutacea (Fr.) Fr. **	x									Russul.	Russul.	Ec
*Russula amoena Quél.*		x	x							Russul.	Russul.	Ec
*Russula amoenicolor Romagn.*	x			x						Russul.	Russul.	Ec
*Russula atropurpurea (Krombh.) Britzelm. **				x						Russul.	Russul.	Ec
*Russula aurea Pers. **	x						x			Russul.	Russul.	Ec
*Russula cavipes Britzelm.*		x								Russul.	Russul.	Ec
*Russula chloroides (Krombh.) Bres.*	x	x		x	x	x	x	x		Russul.	Russul.	Ec
*Russula curtipes F.H. Møller & Jul. Schäff.*	x			x	x		x			Russul.	Russul.	Ec
*Russula delica Fr. **	x	x		x			x	x		Russul.	Russul.	Ec
*Russula densifolia Secr. ex Gillet **	x									Russul.	Russul.	Ec
*Russula faginea Romagn.*	x									Russul.	Russul.	Ec
*Russula foetens Pers. **	x						x			Russul.	Russul.	Ec
*Russula heterophylla (Fr.) Fr. **				x		x	x			Russul.	Russul.	Ec
*Russula illota Romagn.*	x									Russul.	Russul.	Ec
*Russula insignis Quél.*	x				x					Russul.	Russul.	Ec
*Russula luteotacta Rea*	x									Russul.	Russul.	Ec
*Russula nigricans Fr. **	x									Russul.	Russul.	Ec
*Russula nobilis Velen.*	x									Russul.	Russul.	Ec
*Russula ochroleuca Fr. **	x									Russul.	Russul.	Ec
*Russula olivacea (Schaeff.) Fr. **	x									Russul.	Russul.	Ec
*Russula pallidospora J. Blum ex Romagn.*		x								Russul.	Russul.	Ec
*Russula praetervisa Sarnari*			x							Russul.	Russul.	Ec
*Russula risigallina (Batsch) Sacc. **	x					x				Russul.	Russul.	Ec
*Russula romellii Maire*	x									Russul.	Russul.	Ec
*Russula rosea Pers.*	x			x			x			Russul.	Russul.	Ec
*Russula rubroalba (Singer) Romagn.*				x	x		x			Russul.	Russul.	Ec
*Russula sanguinea Fr. **			x							Russul.	Russul.	Ec
*Russula solaris Ferd. & Winge*	x									Russul.	Russul.	Ec
*Russula sororia (Fr.) Romell **					x					Russul.	Russul.	Ec
*Russula torulosa Bres. **			x							Russul.	Russul.	Ec
*Russula vesca Fr. **	x			x						Russul.	Russul.	Ec
*Russula violeipes Quél. **	x									Russul.	Russul.	Ec
*Russula virescens (Schaeff.) Fr. **	x			x			x			Russul.	Russul.	Ec
*Sarcoscypha coccinea (Gray) Boud. **				x						Peziz.	Sarcoscyph.	Sl
*Schizophyllum commune Fr. **	x			x	x	x	x	x		Agaric.	Schizophyll.	Sw
*Scleroderma citrinum Pers. **	x									Bolet.	Sclerodermat.	Ec
*Scleroderma meridionale Demoulin & Malençon*						x	x			Bolet.	Sclerodermat.	Ec
*Scleroderma polyrhizum (J.F. Gmel.) Pers. **				x	x					Bolet.	Sclerodermat.	Ec
*Scleroderma verrucosum (Bull.) Pers. **						x	x	x		Bolet.	Sclerodermat.	Ec
*Scutiger pes-caprae (Pers.) Bondartsev & Singer **	x									Russul.	Albatrell.	Ec
*Stereum hirsutum (Willd.) Pers. Stereaceae*	x						x	x		Russul.	Stere.	Sw
*Strobilomyces strobilaceus (Scop.) Berk. **	x									Bolet.	Bolet.	Ec
*Strobilurus esculentus (Wulfen) Singer*			x							Agaric.	Physalacri.	Sc
*Strobilurus tenacellus (Pers.) Singer*			x							Agaric.	Physalacri.	Sc
*Stropharia aeruginosa (Curtis) Quél. **	x						x			Agaric.	Strophari.	Sl
*Stropharia caerulea Kreisel **	x						x			Agaric.	Strophari.	Sl
*Stropharia rugosoannulata Farl. ex Murrill*		x								Agaric.	Strophari.	Sl
*Suillellus luridus (Schaeff.) Murrill **	x	x		x			x	x		Bolet.	Bolet.	Ec
*Suillellus mendax (Simonini & Vizzini) Vizzini, Simonini & Gelardi*		x								Bolet.	Bolet.	Ec
*Suillellus queletii (Schulzer) Vizzini, Simonini & Gelardi **				x	x	x				Bolet.	Bolet.	Ec
*Suillus bellinii (Inzenga) Kuntze*			x							Bolet.	Suill.	Ec
*Suillus bovinus (L.) Roussel **			x							Bolet.	Suill.	Ec
*Suillus collinitus (Fr.) Kuntze **			x							Bolet.	Suill.	Ec
*Suillus granulatus (L.) Roussel **			x							Bolet.	Suill.	Ec
*Suillus lakei (Murrill) A.H. Sm. & Thiers*			x							Bolet.	Suill.	Ec
*Suillus mediterraneensis (Jacquet. & J. Blum) Redeuilh*			x							Bolet.	Suill.	Ec
*Tapinella atrotomentosa (Batsch) Šutara*		x	x							Bolet.	Tapinell.	Sw
*Tapinella panuoides (Fr.) E.-J. Gilbert*			x							Bolet.	Tapinell.	Sw
*Thaxterogaster purpurascens (Fr.) Niskanen & Liimat. **	x			x	x	x				Agaric.	Cortinari.	Ec
*Thelephora palmata (Scop.) Fr. **		x	x							Thelephor.	Thelephor.	Ec
*Thelephora terrestris Ehrh.*	x			x	x					Thelephor.	Thelephor.	Ec
*Trametes gibbosa (Pers.) Fr. **	x	x								Polypor.	Polypor.	Sw
*Trametes hirsuta (Wulfen) Lloyd **	x			x	x					Polypor.	Polypor.	Sw
*Trametes trogii Berk.*		x								Polypor.	Polypor.	Sw
*Trametes versicolor (L.) Lloyd **	x			x	x	x	x	x		Polypor.	Polypor.	Sw
*Tremella mesenterica Retz. **				x						Tremell.	Tremell.	Sw
*Tricholoma acerbum (Bull.) Quél. **				x						Agaric.	Tricholomat.	Ec
*Tricholoma album (Schaeff.) P. Kumm. **	x			x	x	x	x	x		Agaric.	Tricholomat.	Ec
*Tricholoma argyraceum (Bull.) Gillet*	x									Agaric.	Tricholomat.	Ec
*Tricholoma atrosquamosum Sacc.*	x									Agaric.	Tricholomat.	Ec
*Tricholoma basirubens (Bon) A. Riva & Bon*	x									Agaric.	Tricholomat.	Ec
*Tricholoma columbetta (Fr.) P. Kumm. **	x					x	x			Agaric.	Tricholomat.	Ec
*Tricholoma equestre (L.) P. Kumm. **	x									Agaric.	Tricholomat.	Ec
*Tricholoma quercetorum Contu*				x	x					Agaric.	Tricholomat.	Ec
*Tricholoma saponaceum (Fr.) P. Kumm. **	x			x	x	x	x	x		Agaric.	Tricholomat.	Ec
*Tricholoma scalpturatum (Fr.) Quél.*	x			x	x	x				Agaric.	Tricholomat.	Ec
*Tricholoma sciodes (Pers.) C. Martín*	x									Agaric.	Tricholomat.	Ec
*Tricholoma sejunctum (Sowerby) Quél. **	x			x	x	x	x	x		Agaric.	Tricholomat.	Ec
*Tricholoma sulphureum (Bull.) P. Kumm. **	x						x	x		Agaric.	Tricholomat.	Ec
*Tricholoma terreum (Schaeff.) P. Kumm. **			x							Agaric.	Tricholomat.	Ec
*Tricholoma ustale (Fr.) P. Kumm. **	x			x	x	x	x			Agaric.	Tricholomat.	Ec
*Tricholoma ustaloides Romagn.*				x	x	x	x			Agaric.	Tricholomat.	Ec
*Tricholomopsis rutilans (Schaeff.) Singer **		x	x							Agaric.	Incertae sedis	Sw
*Tuber aestivum Vittad.*					x	x				Peziz.	Tuber.	Ec
*Tuber borchii Vittad. **			x	x	x					Peziz.	Tuber.	Ec
*Tuber brumale Vittad. **	x				x					Peziz.	Tuber.	Ec
*Tuber excavatum Vittad. **	x					x				Peziz.	Tuber.	Ec
*Tuber mesentericum Vittad. **	x		x							Peziz.	Tuber.	Ec
*Tulostoma brumale Pers. **									x	Agaric.	Agaric.	Sm
*Typhula fistulosa (Holmsk.) Olariaga*				x						Agaric.	Typhul.	Sw
*Verpa conica (O.F. Müll.) Sw.*								x		Peziz.	Morchell.	Sl
*Volvariella surrecta (Knapp) Singer*	x									Agaric.	Plute.	Pm
*Volvariella volvacea (Bull.) Singer **	x									Agaric.	Plute.	Sl
*Volvopluteus gloiocephalus (DC.) Vizzini, Contu & Justo **									x	Agaric.	Plute.	St
*Xerocomellus chrysenteron (Bull.) Šutara*	x	x	x							Bolet.	Bolet.	Ec
*Xerocomellus porosporus (Imler exWatling) Šutara*	x									Bolet.	Bolet.	Ec
*Xerocomellus pruinatus (Fr. & Hök) Šutara*	x			x	x	x	x			Bolet.	Bolet.	Ec
*Xerocomus rubellus Quél. **	x				x					Bolet.	Bolet.	Ec
*Xerocomus subtomentosus (L.) Quél. **	x	x	x	x	x	x	x	x		Bolet.	Bolet.	Ec
*Xylaria hypoxylon (L.: Fr.) Grev.*	x	x	x							Xylari.	Xylari.	Sw
*Xylaria polymorpha (Pers.: Fr.) Grev. **	x									Xylari.	Xylari.	Sw

* Taxon recorded in Campania region in [20].

## Data Availability

The original contributions presented in the study are included in the article; further inquiries can be directed to the corresponding author.

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
