# Peer review of "Checklist of Macrofungi Associated with Nine Different Habitats of Taburno-Camposauro Massif in Campania, Southern Italy"

_jof, 2024, doi:10.3390/jof10040275_

Round 1

Reviewer 1 Report

Checklist of Macrofungi associated with nine different habitats of Taburno-Camposauro massif, Campania (South-Italy)

Maurizio Zotti

Checklists provide an informative method for assessing the diversity, geography, and ecology of established and reproducing macrofungi. In addition, given that macrofungi are bioindicator species, their consideration should be included in environmental policies and monitoring of ecosystem conditions.

The author conducted a census of macrofungal species in the Regional Park of Taburno Camposauro (Campania) in nine different habitats between 2019 and 2022. A total of 454 fungal taxa were identified, including several new to the Campania region. Fungal diversity was found to show significant variation depending on the dominant plant species in each habitat. Fagaceae tree species and Carpinus ssp. shared similar fungal communities. Likewise, coniferous tree species had comparable fungal composition. In Abies alba stands and mixed broadleaf forests, a higher prevalence of nonectomycorrhizal taxa was observed, with a concomitant increase in the abundance of saprotrophs. The author found that grasslands recorded lower fungal diversity, suggesting potential effects of wildlife imbalances and overgrazing.

The work is interesting and relevant. The list provided serves as a valuable resource for local governments, offering information for formulating specific management policies. Moreover, authorities in other regions can use this methodology to assess fungal diversity in different areas and understand the ecological status of forest environments.

However, there are a number of questions regarding the work:

Is this collection only for 4 years? Was only one researcher collecting the samples? If the author increases this period to 10-20 years, how might the results of this work change?

In the list of species, agaricoid fungi completely predominate over aphyllophoroid fungi. Does this reflect reality or the author's scientific preferences?

In the list of species, in some places the name of the mushroom family is written immediately after the name of the species, and in others it is not. It needs to be unified.

see above

Author Response

Referee#1

Checklist of Macrofungi associated with nine different habitats of Taburno-Camposauro massif, Campania (South-Italy)

Maurizio Zotti. Checklists provide an informative method for assessing the diversity, geography, and ecology of established and reproducing macrofungi. In addition, given that macrofungi are bioindicator species, their consideration should be included in environmental policies and monitoring of ecosystem conditions.The author conducted a census of macrofungal species in the Regional Park of Taburno Camposauro (Campania) in nine different habitats between 2019 and 2022. A total of 454 fungal taxa were identified, including several new to the Campania region. Fungal diversity was found to show significant variation depending on the dominant plant species in each habitat. Fagaceae tree species and Carpinus ssp. shared similar fungal communities. Likewise, coniferous tree species had comparable fungal composition. In Abies alba stands and mixed broadleaf forests, a higher prevalence of nonectomycorrhizal taxa was observed, with a concomitant increase in the abundance of saprotrophs. The author found that grasslands recorded lower fungal diversity, suggesting potential effects of wildlife imbalances and overgrazing.

The work is interesting and relevant. The list provided serves as a valuable resource for local governments, offering information for formulating specific management policies. Moreover, authorities in other regions can use this methodology to assess fungal diversity in different areas and understand the ecological status of forest environments.

However, there are a number of questions regarding the work:

Thank you for the positive comments.

Is this collection only for 4 years? Was only one researcher collecting the samples? If the author increases this period to 10-20 years, how might the results of this work change?

The collection was carried out from 2019 to 2023, the true period was around 5 years. I erroneously write from 2019 to 2022 in abstract, the correct date is stated in M&M section. In many cases, collection was carried out by my-self and the help of some mycologist of the area, but high number of observations came also from activities of the conservation project “Sve(g)liamo la Dormiente” in which dedicated collection days were carried out in falls 2021 and 2022 and several mycologists were involved. Of course, increasing the period of sampling to 10-20 years should produce a considerable increase in fungal diversity. Unfortunately, I am not able to produce a rarefaction curve describing saturation in number of species and assess if the present dataset is complete or is a partial one. Tendentially, I suppose that is a partial collection of data, as many areas of the regional park were not inspected. Moreover, several undetermined fungi were not included in the checklist because of uncertainty of identifications. Those includes many species belonging to Cortinarius and Russula that are in my advice underestimated. As well as more detailed attention should be dedicated to lignin degrader fungi as resupinate or corticoid basidiomycetes that deserves a more detailed introspection because of their relevant ecological role. A sentence regarding this point has been added to the final paragraph of material discussion section. From line 522 to 524 of track change version: Future research efforts should offer a more comprehensive point of view of macrofungal diversity in Taburno-Camposauro massif compared with the present checklist. Primarily due to challenges in species assessment, especially for underestimated species, as many Cortinarius and Russula; and secondly because of the limited period in which observations were collected, a period of 10-20 years can offer a more detailed view of the fungal diversity in the area.”   

In the list of species, agaricoid fungi completely predominate over aphyllophoroid fungi. Does this reflect reality or the author's scientific preferences?

Yes, Agaricoid fungi predominates, particularly in grassland up to the 86% of the total community. However, in other habitats, Agaricoid species reach maximum values near 50% of the community compared to aphyllophorid ones. I do not know if those levels correspond to the natural ratio between lamellate/aphyllophorid fungi as in many cases checklists of fungi does not provide this information. I suppose that the actual results are due to the size of fungal sporophores as many lamellate fungi are easier to observe during collection days compared to aphyllophorid. However, the question is suggestive for future work comparing variation of lamellate/aphyllophorid ratio in different regions or habitats.

In the list of species, in some places the name of the mushroom family is written immediately after the name of the species, and in others it is not. It needs to be unified.

Done Table has been checked and revised.

Reviewer 2 Report

Dear authors,

This paper is focusing on the Checklist of Macrofungi associated with nine different habitats of Taburno-Camposauro massif, Campania (South-Italy), in which . A total of 454 fungal taxa were identified, including several new records for the Campania region. The fungal diversity exhibited significant variations based on the dominant plant species in each habitat. It can be accepted after major revision.

Please check the revised manuscript with some comments.

Kind Regards

1) To add a paragraph for the Macrofungi diversity in different countries for core areas. To add a paragraph for checklist with ITS sequences to carry out, this can help you to reach the recent researches.

2) To add fungal field pictures for all taxa or core taxa, which are to clear to readers, and it is requirment for all checklist to fungi;

3) The author has to add a table for ITS sequences numbers for all species, which is very important to refer to other mycologists and to make sure the accurate identification for your taxa.

Kind Regards,

Author Response

Referee #2

Dear authors,

This paper is focusing on the Checklist of Macrofungi associated with nine different habitats of Taburno-Camposauro massif, Campania (South-Italy), in which A total of 454 fungal taxa were identified, including several new records for the Campania region. The fungal diversity exhibited significant variations based on the dominant plant species in each habitat. It can be accepted after major revision. Please check the revised manuscript with some comments.

Thank you for the positive comments and the comments in the revised manuscript.

1) To add a paragraph for the Macrofungi diversity in different countries for core areas. To add a paragraph for checklist with ITS sequences to carry out, this can help you to reach the recent researches.

Thank you for the suggestion. However, the present checklist is based on morphological characterization of sporophores collected in Taburno-Camposauro massif. No sequencing has been carried out for specimens collected. This can constitute the basis for future work on the same area. based on this I cannot accomplish other comments regarding the production of ITS number table for each species or the production of phylogenetic trees that however at genus, family and order levels require sequencing of, at least, LSU primers because of the low reliability of ITS at higher taxonomic level. Maybe, the misunderstanding was due to the low level of English that has been revised in the whole text and because in introduction section I mentioned the low reliability of the use of metagenomic techniques in checklists. I am sorry for the inconvenience.

2) To add fungal field pictures for all taxa or core taxa, which are to clear to readers, and it is requirment for all checklist to fungi;

Thank you for the suggestion, I add three panel with pictures of some representative taxa for each habitat included in the survey. I add figure 6, 7 and 8 to the main manuscript.

3) The author has to add a table for ITS sequences numbers for all species, which is very important to refer to other mycologists and to make sure the accurate identification for your taxa.

See response to point 1